# N-terminal cysteine acetylation and oxidation patterns may define protein stability

Karen C. Heathcote [1,2,3], Thomas P. Keeley[2], Matti Myllykoski [4], Malin Lundekvam[4], Nina McTiernan [4], Salma Akter[1], Norma Masson[2], Peter J. Ratcliffe [2,3] ✉, Thomas Arnesen [4,5] ✉ & Emily Flashman [6] ✉

Oxygen homeostasis is maintained in plants and animals by $O_2$-sensing enzymes initiating adaptive responses to low $O_2$ (hypoxia). Recently, the $O_2$-sensitive enzyme ADO was shown to initiate degradation of target proteins RGS4/5 and IL32 via the Cysteine/Arginine N-degron pathway. ADO functions by catalysing oxidation of N-terminal cysteine residues, but despite multiple proteins in the human proteome having an N-terminal cysteine, other endogenous ADO substrates have not yet been identified. This could be because alternative modifications of N-terminal cysteine residues, including acetylation, prevent ADO-catalysed oxidation. Here we investigate the relationship between ADO-catalysed oxidation and NatA-catalysed acetylation of a broad range of protein sequences with N-terminal cysteines. We present evidence that human NatA catalyses N-terminal cysteine acetylation in vitro and in vivo. We then show that sequences downstream of the N-terminal cysteine dictate whether this residue is oxidised or acetylated, with ADO preferring basic and aromatic amino acids and NatA preferring acidic or polar residues. In vitro, the two modifications appear to be mutually exclusive, suggesting that distinct pools of N-terminal cysteine proteins may be acetylated or oxidised. These results reveal the sequence determinants that contribute to N-terminal cysteine protein modifications, with implications for $O_2$-dependent protein stability and the hypoxic response.

Animals and plants have evolved the ability to sense and respond to changes in oxygen availability. Eukaryotes require oxygen for a variety of processes, such as aerobic respiration and metabolism, so low oxygen (hypoxic) conditions lead to metabolic insufficiency and impaired physiological function. One of the most well-studied mechanisms by which animals achieve oxygen homoeostasis is via the hypoxia-inducible factor (HIF) transcription factors[1]. The stability of the HIF transcription factors is regulated by $O_2$-sensitive HIF prolyl

hydroxylase (PHD) enzymes[2]. In normoxia, hydroxylation of HIFα domains can be catalysed by PHDs, leading to HIF degradation[3,4]. In hypoxia, however, reduced oxygen availability abrogates the activity of PHD enzymes. Stabilised HIF transcription factors are able to induce a transcriptional response through binding to hypoxia response elements (HREs) in target genes. These genes regulate a variety of processes, including metabolism, angiogenesis, and ventilation, to help decrease $O_2$ consumption and increase $O_2$ delivery[5]. In plants, a similar

[1]Department of Chemistry, University of Oxford, OX1 3TA Oxford, UK. [2]Ludwig Institute for Cancer Research, Nuffield Department of Medicine, University of Oxford, OX3 7FZ Oxford, UK. [3]The Francis Crick Institute, 1 Midland Road, NW1 1AT London, UK. [4]Department of Biomedicine, University of Bergen, 5020 Bergen, Norway. [5]Department of Surgery, Haukeland University Hospital, 5021 Bergen, Norway. [6]Department of Biology, University of Oxford, OX1 3RB Oxford, UK. ✉e-mail: peter.ratcliffe@ndm.ox.ac.uk; thomas.arnesen@uib.no; emily.flashman@biology.ox.ac.uk

system exists: the stability of transcription factors called Group VII ethylene response factors (ERF-VIIs) is regulated by $O_2$-sensitive plant cysteine oxidase (PCO) enzymes[6]. The PCO enzymes catalyse oxidation of the N-terminal cysteine (Nt-Cys) of the ERF-VIIs, which triggers their degradation via the Cys/Arg N-degron pathway[7,8]. As with the PHD enzymes, hypoxia leads to reduced PCO activity, stabilising ERF-VII transcription factors, which can subsequently induce the expression of hypoxia response genes[9,10]. Hypoxia response genes then regulate processes that increase supply or reduce demand for $O_2$, allowing plants to adapt to hypoxic conditions[11,12].

2-Aminoethanethiol dioxygenase (ADO, also known as cysteamine dioxygenase) is a PCO-like enzyme present in humans[13]. Similar to the PCOs, human ADO (HsADO) was shown to regulate the stability of proteins in an $O_2$-dependent manner via the Cys/Arg N-degron pathway (Fig. 1A)[13]. Three protein substrates of HsADO were characterised: regulators of G-protein signalling 4 and 5 (RGS4/5), and chemokine interleukin-32 (IL32), whose stability were ADO- and $O_2$-dependent[13,14]. Interestingly, HsADO was found to have similar oxygen sensitivity to the PHD enzymes that regulate the HIF transcriptional response. A $K_mO_2^{app}$ of approximately 50% $O_2$ was calculated for HsADO with RGS4 and RGS5, compared to > 30% (> 400 µM) for PHD2 with HIF[13,15]. So, both regulatory enzymes have a $K_mO_2$ above the physiological range and are, therefore, highly sensitive to changes in $O_2$ levels. ADO-mediated control of protein stability therefore has the potential to be a significant regulator of physiological responses to hypoxia, in particular to provide a rapid response to hypoxia, compared to regulation via the HIF system which initiates hypoxic adaption by increasing gene expression[13].

RGS4, RGS5 and IL32 are not the only human proteins to possess the Nt-Cys motif. According to UniProt[16], there are approximately 200–300 potential Met-Cys-initiating (MC-initiating) sequences in the human proteome (Supplementary Data 1). These represent a broad range of potential HsADO substrates that could be hypoxically regulated. The question therefore arises as to how many Nt-Cys proteins are substrates of HsADO. We sought to investigate the scope of HsADO's role in regulating $O_2$-dependent protein stability by screening a variety of MC-initiating sequences from the human proteome as substrates both in vitro and in cells. HsADO was found to be selective among Nt-Cys sequences in both peptide activity and cell-reporter assays, with a preference for basic or aromatic residues at the position following the Nt-Cys. Notably, we found that some sequences were unexpectedly not regulated by ADO in cell-reporter assays, raising the

possibility that their N-termini were protected from oxidation. Given that cross-talk between different post-translational modifications on the same residues has been reported to dictate protein fate, e.g. acetylation/ubiquitination of Lys residues and phosphorylation/O-GlcNAcylation of Ser/Thr residues[17,18], we considered whether other N-terminal modifications could be influencing ADO activity in cells.

There exists a broad range of protein N-terminal modifications including acetylation, myristoylation and initiator methionine cleavage[19]. Initiator methionine (iMet) cleavage by methionine aminopeptidases (MetAPs) occurs co-translationally to a high stoichiometry when iMet is followed by a small amino acid (Cys, Ser, Thr etc.) thus creating neo-N-termini (Nt-Cys, Nt-Ser, Nt-Thr etc.) and usually precedes other modifications[20]. One of the most common N-terminal modifications is acetylation, which can lead to a range of functional effects for the target protein, including stabilisation[21–23]. We therefore chose to focus on Nt-acetylation, which is catalysed by N-terminal acetyltransferases (NATs), with NatA being the enzyme presumed to be responsible for Nt-Cys acetylation (Fig. 1B)[24]. Our results provided definitive evidence that human NatA is responsible for Nt-Cys acetylation. Interestingly, NatA was also found to be selective among Nt-Cys sequences in vitro, with a preference for sequences with acidic or polar residues immediately following the Nt-Cys, distinct from the substrate preferences of ADO. This suggested that NatA and ADO have evolved to modify distinct pools of Nt-Cys-initiating proteins, which has implications for the subsequent fates of these different substrates. Despite these broadly distinct substrate preferences, we did identify some Nt-Cys initiating sequences with the potential to be substrates of either ADO or NatA, and showed that certain ADO and NatA substrate peptides could be shielded from the opposing modification in vitro. While low levels of enzymatic activity towards these substrates made it challenging to verify in vivo, this observation could rationalise some of the conflicting outcomes for ADO oxidation assays between peptide- and cell-based assays and may dictate the pool of substrates available for ADO and $O_2$-dependent proteostatic regulation.

## Results and discussion
### ADO shows substrate selectivity beyond Nt-Cys in vitro
We first sought to establish the extent and determinants of the substrates of HsADO (henceforth ADO) within the human proteome. This was initially addressed using in vitro peptide activity assays, whereby the ADO-catalysed oxidation of a range of Nt-Cys initiating sequences was

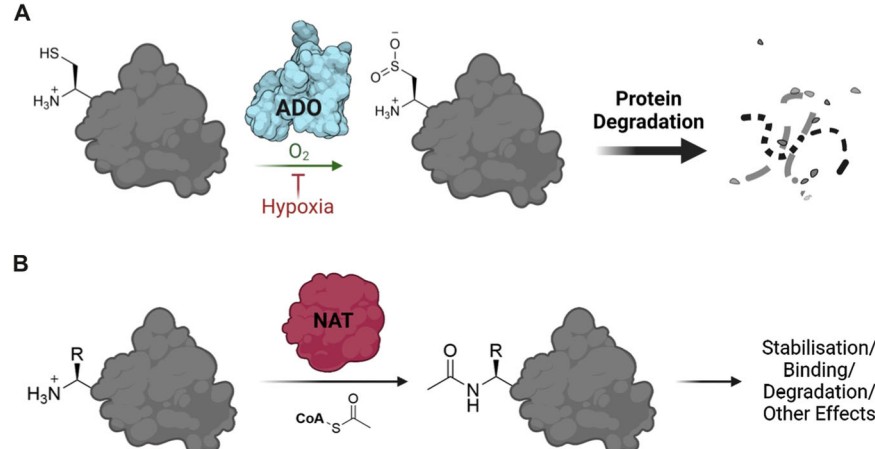

**Fig. 1 | Oxidation and acetylation of protein N-termini. A** Cysteamine dioxygenase (ADO) can catalyse the oxidation of the N-terminal cysteine (Nt-Cys) of a target protein, incorporating two oxygen atoms from molecular oxygen[13,49]. This allows the protein to be recognised and so degraded by the Arg/N-degron pathway[50]. ADO-catalysed Nt-Cys oxidation is significantly reduced in hypoxia. **B** N-terminal acetyltransferase enzymes (NATs) catalyse the acetylation of the N-terminal residue of a target protein, involving the transfer of an acetyl group from acetyl coenzyme A to the N-terminal amine. This can have a range of effects including stabilisation of the target protein, enhanced binding and degradation by the acetyl branch of the N-degron pathway[24]. Created using Biorender (license agreement #YB26TL2XO9).

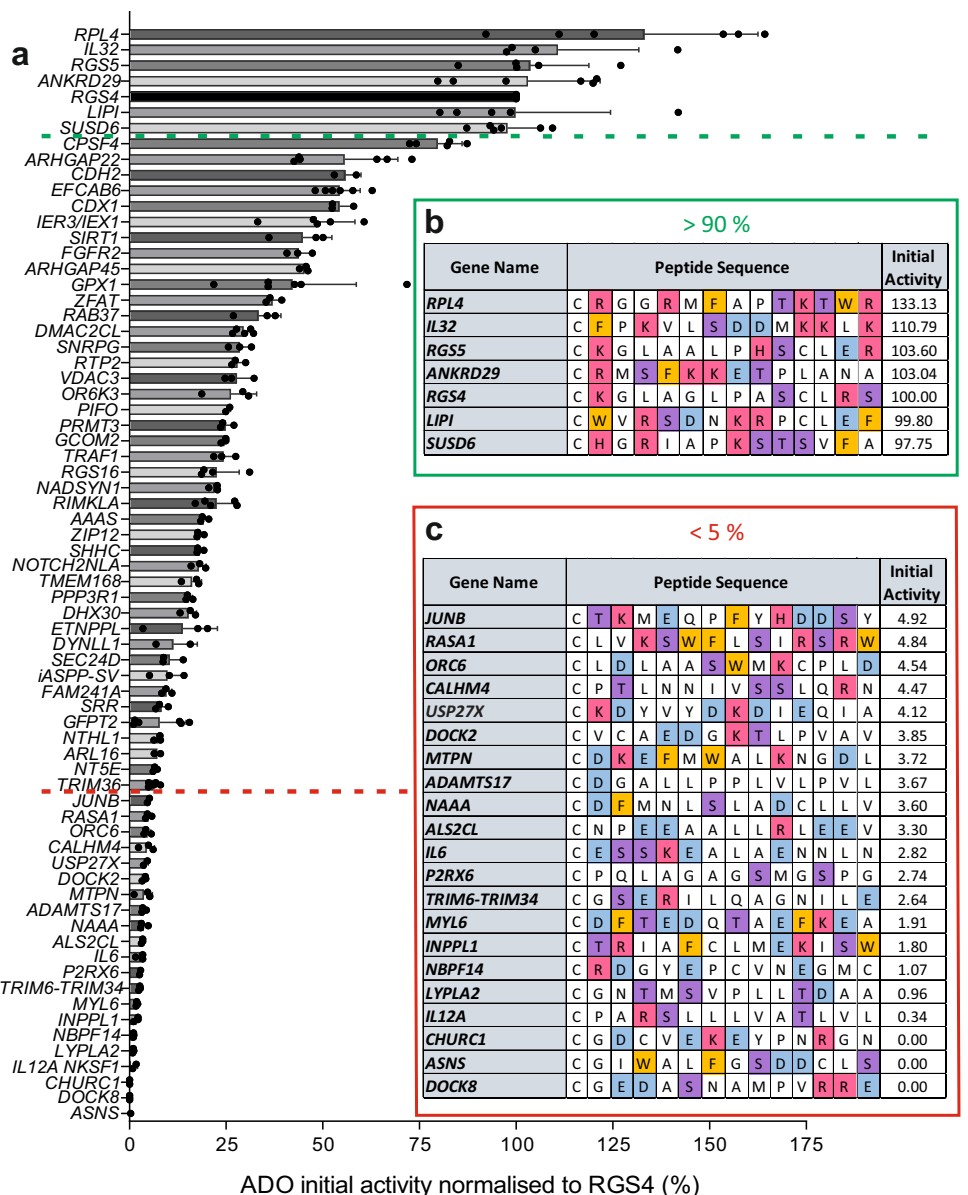

**Fig. 2 | In vitro screening of Nt-Cys initiating peptide sequences from the human proteome with recombinant *Hs*ADO. a** Bar graphs show ADO-catalysed oxidation of 14-mer peptides representing the (Met-cleaved) Cys-initiating N-termini of potential ADO substrates (listed by gene name) after 1 min; activity was normalised to oxidation of RGS4$_{2\text{-}15}$ positive control peptide. Data are presented as mean ± standard deviation, $n \geq 3$. **b, c** Sequences of peptides with which ADO was observed to have > 90% (**b**) or < 5% (**v**) activity relative to RGS4$_{2\text{-}15}$ positive control peptide. Amino acids colour coded according to side chain chemical properties: basic (H/K/R) = red; acidic (D/E) = blue; aromatic (F/W) = orange; polar (S/T) = purple. Source data are provided as a Source Data file.

measured using peptides representing residues 2–15 of MetCys-initiating proteins (from which iMet residues have been excised by MetAPs[20]). A peptide representing the known substrate RGS4[13] was included as a positive control in all assays. We selected 68 additional MC-initiating sequences from the human proteome to screen as potential ADO substrates. These sequences were selected to represent a diversity in the identity of the amino acid at the position following the Nt-Cys, including some that were similar to established substrates[13], which could provide information on ADO's substrate preferences and the potential of their corresponding proteins to be regulated by ADO in vivo.

Peptides representing the Met-excised N-termini of these proteins were incubated with recombinant ADO for 1 min at 37 °C (Fig. 2, Supplementary Data 2) to measure initial activity. Mass spectrometry was used to quantify oxidation and dioxygenation of the peptides was characterised by a + 32 Da mass addition (Supplementary Fig. 1). Results were normalised within each assay to oxidation of the RGS4$_{2\text{-}15}$

peptide. The level of oxidation of the Nt-Cys peptides tested in vitro varied from 0% to > 130% relative to RGS4$_{2\text{-}15}$ (Fig. 2). The most rapidly oxidised peptide from this initial screen represented the N-terminus of a computationally mapped non-canonical isoform of 60 S ribosomal protein L4 (accession H3BM89, RPL4). Other N-terminal sequences, including those of ankyrin repeat domain-containing protein 29 (ANKRD29) and sushi domain-containing protein 6 (SUSD6), were oxidised to a level comparable to RGS4 (Fig. 2a, b). Known substrates RGS5 and IL32 were also included in in vitro screening assays and were similarly active to RGS4. Meanwhile, some peptides showed no detectable oxidation under these conditions: glutamine-dependent asparagine synthetase (ASNS), protein Churchill (CHURC1), and a (now obsolete) isoform of dedicator of cytokinesis 8 (DOCK8). ADO oxidised other peptides at a range of activities between these extremes.

The variability in the peptide activity assay data indicated that ADO shows selectivity in vitro towards certain Nt-Cys sequences and

## a

| Name | 2 | 3 | 4 | 5 | 6 | 7 | 8 | 9 | 10 | 11 | 12 | 13 | 14 | 15 | Initial Activity |
|---|---|---|---|---|---|---|---|---|---|---|---|---|---|---|---|
| *Residue* | 2 | 3 | 4 | 5 | 6 | 7 | 8 | 9 | 10 | 11 | 12 | 13 | 14 | 15 | |
| RGS4 | C | K | G | L | A | G | L | P | A | S | C | L | R | S | 100.00 |
| RGS5 | C | K | G | L | A | A | L | P | H | S | C | L | E | R | 103.60 |
| IL32 | C | F | P | K | V | L | S | D | D | M | K | K | L | K | 110.79 |
| OS1 | C | R | G | R | K | K | A | T | T | K | A | W | R | | 81.17 |
| OS2 | C | R | G | G | R | K | K | A | T | T | K | A | W | R | 77.15 |
| OS3 | C | R | G | G | F | K | K | A | T | T | K | A | W | R | 120.78 |
| OS4 | C | R | G | G | F | M | K | A | T | T | K | A | W | R | 99.56 |
| OS5 | C | R | G | G | F | M | K | A | P | T | K | A | W | R | 100.79 |
| OS6 | C | K | G | R | M | K | A | T | T | K | A | W | R | | 68.73 |
| OS7 | C | F | G | R | M | K | A | T | T | K | A | W | R | | 152.16 |
| OS8 | C | R | M | R | M | K | A | T | T | K | A | W | R | | 135.14 |
| OS9 | C | R | P | R | M | K | A | T | T | K | A | W | R | | 116.27 |
| OS10 | C | R | G | R | M | K | A | T | T | K | A | W | R | | 112.65 |

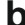
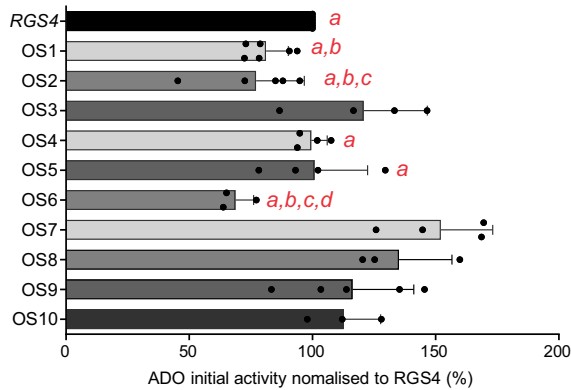

## b

| Name | 2 | 3 | 4 | 5 | 6 | 7 | 8 | 9 | 10 | 11 | 12 | 13 | 14 | 15 | Initial Activity |
|---|---|---|---|---|---|---|---|---|---|---|---|---|---|---|---|
| *Residue* | 2 | 3 | 4 | 5 | 6 | 7 | 8 | 9 | 10 | 11 | 12 | 13 | 14 | 15 | |
| RGS4 | C | K | G | L | A | G | L | P | A | S | C | L | R | S | 100.00 |
| ASNS | C | G | I | W | A | L | F | G | S | D | D | C | L | S | 0.01 |
| RGS4 K3D | C | D | G | L | A | G | L | P | A | S | C | L | R | S | 12.03 |
| RGS4 K3G | C | G | G | L | A | G | L | P | A | S | C | L | R | S | 9.33 |
| AS1 | C | G | E | L | A | M | K | A | T | T | K | A | W | R | 2.88 |
| AS2 | C | G | S | L | A | M | K | A | T | T | K | A | W | R | 31.56 |
| AS3 | C | D | E | L | A | M | K | A | T | T | K | A | W | R | 2.67 |
| AS4 | C | D | S | L | A | M | K | A | T | T | K | A | W | R | 3.17 |
| AS5 | C | T | E | L | A | M | K | A | T | T | K | A | W | R | 13.63 |
| AS6 | C | T | S | L | A | M | K | A | T | T | K | A | W | R | 30.50 |
| AS7 | C | G | E | L | A | M | P | A | S | E | E | S | L | S | 0.00 |
| AS8 | C | G | S | L | A | M | P | A | S | E | E | S | L | S | 0.00 |
| AS9 | C | D | E | L | A | M | P | A | S | E | E | S | L | S | 0.00 |
| AS10 | C | D | S | L | A | M | P | A | S | E | E | S | L | S | 0.00 |
| AS11 | C | T | E | L | A | M | P | A | S | E | E | S | L | S | 0.00 |
| AS12 | C | T | S | L | A | M | P | A | S | E | E | S | L | S | 2.89 |

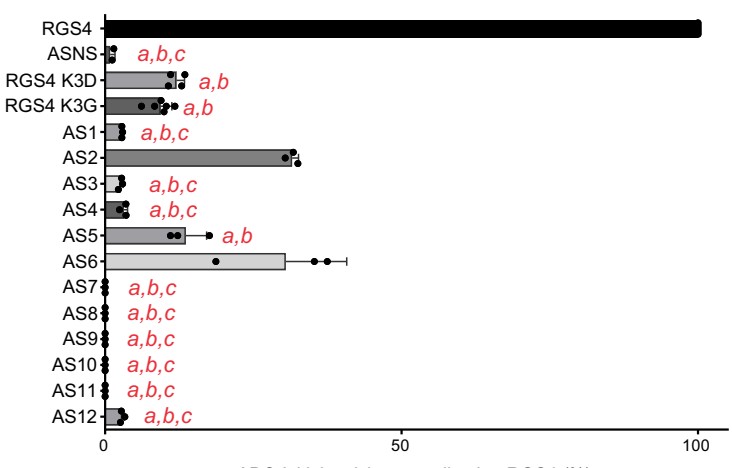

**Fig. 3 | In vitro testing of proposed ADO optimum substrate (OS) and anti-substrate (AS) peptides. a** Sequences of ADO optimum substrate (OS) peptides and initial activity of *Hs*ADO with OS peptides. Significant differences ($p < 0.05$) from 1-way ANOVA results summarised on graph: $a$ = lower than OS7; $b$ = lower than OS8; $c$ = lower than OS3; $d$ = lower than OS9 and OS10. Full 1-way ANOVA results and *p*-value ranges available in Supplementary Data 3. **b** Sequences of ADO anti-substrate (AS) peptides and initial activity of *Hs*ADO with AS peptides and RGS4 K3D/G. ASNS was included as a negative control for ADO activity. Significant differences ($p < 0.05$) from 1-way ANOVA results summarised on graph: $a$ = lower than AS2; $b$ = lower than AS6; $c$ = lower than AS5; ASNS, RGS4 K3D/G and AS1-12 all lower than RGS4. Full 1-way ANOVA results available and *p*-value ranges in Supplementary Data 4. For (**a**, **b**), initial activity is turnover of peptide after 1-min incubation with *Hs*ADO normalised to that of RGS4$_{2-15}$ positive control peptide (%). Data represent mean ± standard deviation; $n \geq 3$. Amino acids are colour coded according to chemical properties: basic (H/K/R) = red; acidic (D/E) = blue; aromatic (F/W) = orange; polar (S/T) = purple. Source data are provided as a Source Data file.

also provided information on the favourable amino acid composition of substrates. Peptides towards which ADO showed the highest rate of activity (> 90% relative to RGS4) all had a basic (Lys/Arg/His) or aromatic (Phe/Trp) residue after the Nt-Cys (Fig. 2b). Conversely, sequences towards which ADO showed less than 5% activity relative to RGS4 had a higher prevalence of acidic residues (Fig. 2c).

To test ADO's emerging substrate preferences, we designed ten 'optimum substrate' (OS) and twelve 'anti-substrate' (AS) peptides based on the Nt-sequences of the best (Fig. 2b) and worst (Fig. 2c) substrates identified in the initial screening. We also replaced the basic Lys residue at position 3 in RGS4 with an acidic Asp or neutral Gly to directly test the impact of this residue in substrate selectivity (RGS4 K3D/G). OS peptides contained high proportions of basic and aromatic

residues and, as anticipated, were all rapidly oxidised by ADO (Fig. 3a, Supplementary Data 3); the lowest activity relative to RGS4 was observed towards OS6 (69%) and the highest OS7 (152%). OS6, OS7 and OS10 (112%) sequences only differ at the residue following Nt-Cys (Lys, Phe and Arg, respectively), suggesting that small changes at this position can significantly influence ADO substrate preferences. The most active OS peptide (OS7) shares a sequence motif with two of the best substrates from initial in vitro screening (IL32 and a computationally mapped isoform of lipase member I (LIPI) (Fig. 2a, b)): an aromatic amino acid at residue 3 followed by a basic amino acid at residue 5. AS peptides contained acidic and/or polar residues at residues 3 and 4 and, as anticipated, were predominantly weak substrates of ADO (Fig. 3b, Supplementary Data 4). Low levels of oxidation were

## a

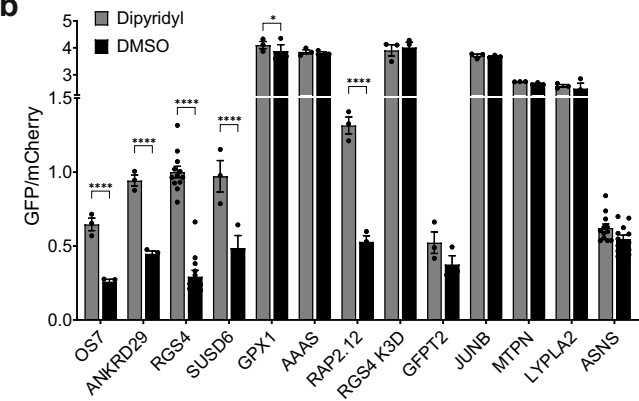

| Name | N-terminal Sequence | | | | | | | | | | | | | | in vitro Activity |
|---|---|---|---|---|---|---|---|---|---|---|---|---|---|---|---|
| OS7 | M | C | F | G | R | R | M | K | A | T | T | K | A | W | R | 152.16 |
| ANKRD29 | M | C | R | M | S | F | K | K | E | T | P | L | A | N | A | 103.04 |
| RGS4 | M | C | K | G | L | A | G | L | P | A | S | C | L | R | S | 100.00 |
| SUSD6 | M | C | H | G | R | I | A | P | K | S | T | S | V | F | A | 97.75 |
| GPX1 | M | C | A | A | R | L | A | A | A | A | A | A | A | Q | S | 42.05 |
| AAAS | M | C | S | L | G | L | F | P | P | P | P | R | G | G | Q | 19.24 |
| At RAP2.12 | M | C | G | G | A | I | I | S | D | F | I | P | P | P | R | 16.36 |
| RGS4 K3D | M | C | D | G | L | A | G | L | P | A | S | C | L | R | S | 12.03 |
| GFPT2 | M | C | G | I | F | A | Y | M | N | Y | R | V | P | R | T | 7.78 |
| JUNB | M | C | T | K | M | E | Q | P | F | Y | H | D | D | S | Y | 4.92 |
| MTPN | M | C | D | K | E | F | M | W | A | L | K | N | G | D | L | 3.72 |
| LYPLA2 | M | C | G | N | T | M | S | V | P | L | L | T | D | A | A | 0.96 |
| ASNS | M | C | G | I | W | A | L | F | G | S | D | D | C | L | S | 0.00 |

## b

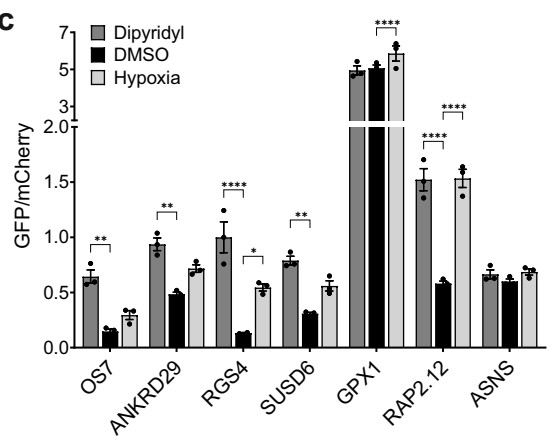

## c

**Fig. 4 | Regulation of potential ADO substrate sequences in Dual-Fluorescence Oxygen Reporter (DFOR) assay. a** Sequences of proteins selected from in vitro screening for testing in cells; including RAP2.12, a substrate of related Plant Cysteine Oxidase (PCO) enzymes. Amino acids are colour coded according to chemical properties: basic (H/K/R) = red; acidic (D/E) = blue; aromatic (F/W) = orange; polar (S/T) = purple. "in vitro Activity" values are relative to RGS4$_{2-15}$ (%). **b**, **c** SH-SY5Y cell lines stably expressing MCX$_{13}$GFP:P2A:mCherry were (**b**) treated for 20 h with 2,2-dipyridyl (200 μM) or DMSO vehicle control and (**c**) incubated for 24 h in normoxia or hypoxia (1% O$_2$). GFP/mCherry ratio was calculated using fluorescence measurements and normalised to the GFP:mCherry ratio obtained in cells expressing the positive control construct RGS4$_{(1-15)}$-GFP:P2A:mCherry. Higher GFP/mCherry ratio indicates stabilisation of eGFP protein; increased stability after dipyridyl or hypoxia treatment is consistent with reduced ADO activity. Error bars show SEM. Statistical significance determined with 2-way ANOVA with Holm-Sidak multiple comparisons post-hoc test: ns ($p > 0.05$); *$p \leq 0.05$; **$p \leq 0.01$; ***$p \leq 0.001$; ****$p \leq 0.0001$. Data were collected across at least three biological repeats. Source data are provided as a Source Data file.

also seen in RGS4 K3D/G, indicating the importance of a bulky or basic residue in this position. Peptides AS1-6 shared C-terminal residues 8-15 (KATTKAWR) with the OS peptides and were oxidised to low or medium levels by ADO (~2–30% compared to RGS4$_{2-15}$); low levels of oxidation were observed in substrates with acidic residues at positions 3 and 4. Peptides AS7-12 had C-terminal residues (PASEESLS) comprising predominantly polar and negatively charged residues; no oxidation of AS7-11 was observed on the 1-min timescale, and oxidation of these peptides remained very low even after 30 min (Supplementary Fig. 3). AS12, with two polar residues instead of acidic residues at positions 3 and 4) was oxidised to a low level (2.89%); comparison of peptides AS12 and AS6 (30.5%), which are identical in residues 2-7 but differing in their C-terminal residues, indicates that residues that are substantially downstream of the Nt-Cys can significantly impact on ADO activity. Overall, these assays (Fig. 3, Supplementary Fig. 2) confirmed the preference of ADO for peptide substrates with basic residues and the negative impact of acidic residues.

To investigate ADO's structure-activity relationship in relation to substrate selectivity, we used a molecular modelling approach, performing docking experiments with MmADO (7LVZ, [https://www.rcsb. org/structure/7LVZ] (crystal structure of MmADO)[25] and two 7-mer substrate peptides (RGS4$_{2-8}$ and IL32$_{2-8}$) using AutoDock4 (Supplementary Fig. 4). Fernandez et al. conducted docking and QM/MM optimisation studies with a 4-mer RGS5$_{2-5}$ peptide (which shares sequence identity with RGS4$_{2-5}$) and noted a potential interaction between the N-terminal amine group of RGS5 and Asp192[25]. Our docking results using longer peptides also suggest that Asp192 could interact with this N-terminal amine (Supplementary Fig. 4a–c). In addition, we identified a 'polar pocket' in the active site consisting of Tyr75, Glu80, and Ser85 (HsADO equivalent residues: Tyr87, Glu92, Ser97), which is well positioned to bind the side chains of basic residues at position 3 in a substrate, e.g., Lys in RGS4/5 (Supplementary Fig. 4). Docking experiments with IL32 show that ADO's active site cavity is open enough to accommodate bulky aromatic side chains at residue 3 in a substrate, while still retaining an interaction between the Nt-amine and Asp192 (Supplementary Fig. 4b). We found that these modelling experiments supported observations from peptide activity assays regarding ADO's substrate preferences. The docking results also suggested that substrate residues beyond position 7 may be able to interact with ADO residues in a potential substrate-binding hairpin loop[25,26], rationalising the contribution this region appeared to make to ADO activity towards AS peptides.

### Substrate selectivity of HsADO is also observed in cells

Peptide activity assays in vitro clearly show that ADO prefers substrates with basic and aromatic residues at the N-terminus, while acidic residues are unfavourable. However, it is important to confirm that this preference is also observed in a cellular context. We used a dual-fluorescence oxygen reporter (DFOR) assay[27] (Supplementary Fig. 5) to determine how in vitro activity of ADO on specific peptides related to the ability of these peptides to confer iron and oxygen-regulated stability on an intracellular reporter protein, as might be predicted from the action of ADO.

In this DFOR assay, we fused the N-terminal residues of the relevant protein to the N-terminus of GFP, allowing us to monitor protein stability by fluorescence and so investigate N-terminal sequence preferences in cells (the rationale of this assay is described in detail in ref. 27 and Supplementary Fig. 5). Eleven sequences were selected from the in vitro screening assays for further investigation in cells (Fig. 4a), based on their range of in vitro activities (0–103% compared to RGS4). In addition, we included RGS4 K3D, OS7 (the most active OS peptide, Fig. 3a) and RAP2.12$_{2-15}$, a peptide representing the Cys-initiating N-terminus of an ERF-VII transcription factor whose stability is regulated by PCOs in Arabidopsis thaliana[6,8] to determine whether they could also confer ADO-dependent protein stability.

We produced SH-SY5Y cell lines stably expressing MCX$_{13}$-GFP:P2A:mCherry constructs, where MCX$_{13}$ corresponded to the N-terminal sequence of each protein of interest (Fig. 4a, Supplementary Fig. 5) (and thus to the peptides used in the biochemical assays (Fig. 2)). The constructs used included mCherry separated from the modified GFP by a self-cleaving P2A sequence; ensuring that the GFP and mCherry proteins are expressed at proportional levels within the cell and that any reduction in the GFP:mCherry ratio therefore reflects a change in the stability of the GFP protein. ADO activity in cells was inhibited by treatment with 2,2-dipyridyl or hypoxia, both of which have previously been shown to inhibit ADO activity in vivo[13].

Initially, we treated cells expressing all constructs with 2,2-dipyridyl (200 μM) or DMSO vehicle control and then incubated them in normoxia for 24 h. Conjugation of the N-terminus of RGS4, a known in vivo substrate of ADO[13], to GFP resulted in a > 3-fold average increase in GFP:mCherry ratio after 2,2-dipyridyl treatment compared to DMSO control (Fig. 4b, Supplementary Data 5), reflecting activity-dependent destabilisation by ADO. When the sequences of some of the most active in vitro substrates (OS7, ANKRD29, and SUSD6 (Fig. 4a)) were fused to GFP, 2,2-dipyridyl treatment similarly led to a > 2-fold increase in GFP:mCherry ratio (Fig. 4b, Supplementary Data 5). GFP fusion protein stabilisation was also observed with the RAP2.12 Nt-sequence (2.5-fold increase with 2,2-dipyridyl Fig. 4b). These data are consistent with these N-terminal sequences also acting as ADO substrates in cells. Consistent with in vitro data (Fig. 4a), GFP stability did not appear to be significantly impacted by ADO function when preceded by the Nt-Cys initiating sequences of glutamine-fructose-6-phosphate aminotransferase 2 (GFPT2), transcription factor JunB (JUNB), myotrophin (MTPN), acyl-protein thioesterase 2 (LYPLA2), ASNS or indeed the RGS4 K3D variant (Fig. 4b), indicating they were poor ADO substrates both in vitro and in cells. Interestingly, both Aladin (AAAS) and glutathione peroxidase 1 (GPX1) sequences were not regulated in cells using the DFOR assay, despite higher ADO activity being observed towards these peptides in vitro (19% and 42%, respectively, relative to RGS4).

In further experiments, the GFP-fusions that were regulated by 2,2-dipyridyl, i.e., were shown to be ADO substrates in cells, were also tested with respect to the effects of hypoxia on stabilisation (Fig. 4c). Slight stabilisation was observed for most sequences, though this was only statistically significant for RGS4$_{1-15}$-GFP and RAP2.12$_{1-15}$-GFP. ASNS- and GPX1-GFP fusions were also tested for the effects of hypoxia; as expected, these were not regulated by ADO in this context, although a small degree of GPX1$_{1-15}$-GFP stabilisation was seen following hypoxia treatment (1.2-fold increase).

Overall, these data show that ADO continues to be selective amongst Nt-Cys sequences in a cellular context and, for those that were ADO substrates, destabilisation was O$_2$-dependent. While most DFOR assay results were consistent with in vitro activity assays, the inability of AAAS and GPX1 Nt-sequences to confer the predicted ADO substrate properties onto GFP were exceptions to this observation. This led us to hypothesise that a competing N-terminal Cys modification could be influencing ADO's activity towards these substrates in cells.

## NatA acetylates Nt-Cys and shows substrate selectivity beyond Nt-Cys

N-terminal acetylation, catalysed by N-terminal acetyltransferase (NAT) enzymes, is one of the most common modifications in eukaryotes[28]. N-amino acetylation of an N-terminal Cys residue would likely prevent binding to ADO, in particular via its proposed interaction with Asp192 in the active site[25]. Acetylation of small N-terminal amino acids following methionine cleavage is catalysed by N-terminal acetyltransferase A (NatA) in human cells[28,29]. Evidence for the specific activity of NatA towards Nt-Cys is, however, lacking. Obtaining direct cellular evidence of NatA-mediated Nt-Cys acetylation is challenging since the genes encoding NatA's catalytic subunit, NAA10, and its

ribosome binding subunit, NAA15, are essential genes[30,31] and human KO cells are not available. Furthermore, knockdown of NatA subunits NAA10 and NAA15 only partially reduces acetylation levels of a fraction of the true cellular NatA substrate pool and Nt-Cys proteins are rarely identified[28]. To verify that NatA is responsible for the cellular Nt-acetylation of Nt-Cys proteins and to enable efficient substrate retrieval in a human HAP1 cell line, we first combined NAA10 knockdown with overexpression and subsequent immunoprecipitation of JUNB, a protein known to be Nt-acetylated in vivo. HAP1 ADO KO cells were used to avoid potential ADO-mediated oxidation of JUNB. Cells were treated with control siRNA (siCtrl), siNAA10, or siNAA20 as a NAT-specificity control (Fig. 5a). NAA20 is the catalytic subunit of the human NatB complex acetylating a subgroup of Met-starting N-termini and is not predicted to acetylate Nt-Cys proteins[32,33]. Expression of JUNB-Flag was followed by anti-Flag immunoprecipitation and mass spectrometry. We identified N-terminal peptides representing Nt-acetylated JUNB in all conditions, while non-Nt-acetylated JUNB peptides were only identified in siNAA10 cells (Fig. 5b, c, Supplementary Fig. 6). Thus, knockdown of NatA shifts cellular JUNB from a fully to a partially Nt-acetylated state. In addition, we used budding yeast strains naa10Δ and naa20Δ, fully lacking functional NatA and NatB, respectively. These were transformed with a JUNB-FLAG plasmid or control plasmid and subjected to immunoprecipitation and LC-MS/MS (Fig. 5d, e, Supplementary Fig. 7). N-terminal peptides representing Nt-acetylated JUNB were only identified in cells containing NatA, while non-Nt-acetylated JUNB peptides were identified in all yeast strains. These experiments allowed complete removal of NatA-associated acetylation and, collectively, definitively demonstrate NatA as responsible for Nt-Cys acetylation.

We next, therefore, tested a selection of the MC-initiating proteins, previously screened against HsADO in vitro (Fig. 2), as potential substrates of HsNatA in vitro using a radioactivity-based assay[34]. Peptides with the sequence CX$_6$-RWGRPVGRRRRPVRVYP (where CX$_6$ was the first 7 amino acids of the Met-excised N-terminal sequence of a protein of interest) were incubated with recombinant NatA protein and a 1:3 mix of [$^{14}$C]-labelled and unlabelled acetyl CoA for 5 min at 37 °C. The amino acid sequence following the CX$_6$ residues mediated binding of peptides to the filter disks during extraction for analysis. Activity was measured by calculating [$^{14}$C]-acetyl incorporation into the peptide, which we normalised to [$^{14}$C]-acetyl incorporation into a SESSSKSRWGRPVGRRRRPVRVYP positive control peptide derived from the in vivo fully Nt-acetylated NatA substrate high mobility group protein HMG-I(Y) (HMGA1)[28].

As seen for ADO, the activity of NatA towards the sequences tested showed a high degree of variation, ranging from 0% to 120% compared with the HMGA1 positive control peptide (Fig. 6a). Interestingly, this selectivity appeared to be based on the physico-chemical properties of the amino acids downstream of the Nt-Cys residue, but with very different selectivity determinants compared to those of ADO. The highest activity of NatA was observed towards sequences that had acidic (Asp/Glu) or polar (Ser/Thr) amino acids at the position following the Nt-Cys, including AAAS, MTPN and JUNB (Fig. 6b). Conversely, NatA had very low activity (< 1% relative to HMGA1) towards sequences with basic or aromatic residues after the Nt-Cys-, such as OS7, ANKRD29 and RGS4 (Fig. 6b, Supplementary Table 1). To probe the substrate preferences of NatA further, a series of MTPN-D3X mutant peptides were tested with NatA (Fig. 6b). In these mutants the Asp residue following the Nt-Cys (D3) in MTPN was mutated to a polar (Ser, S), basic (Arg, R), hydrophobic (Gly, G; Leu, L) or alternative acidic (Glu, E) residue. NatA activity was almost completely lost when D3 was mutated to a basic residue, dropping from 22.55% with the wild-type to 0.51% with the D3R mutant. Activity was also greatly reduced for hydrophobic D3L and D3G mutants, but was partially retained when Asp was mutated to acidic Glu and polar Ser. We also tested NatA activity towards the mutated form of RGS4, K3D; here, NatA activity

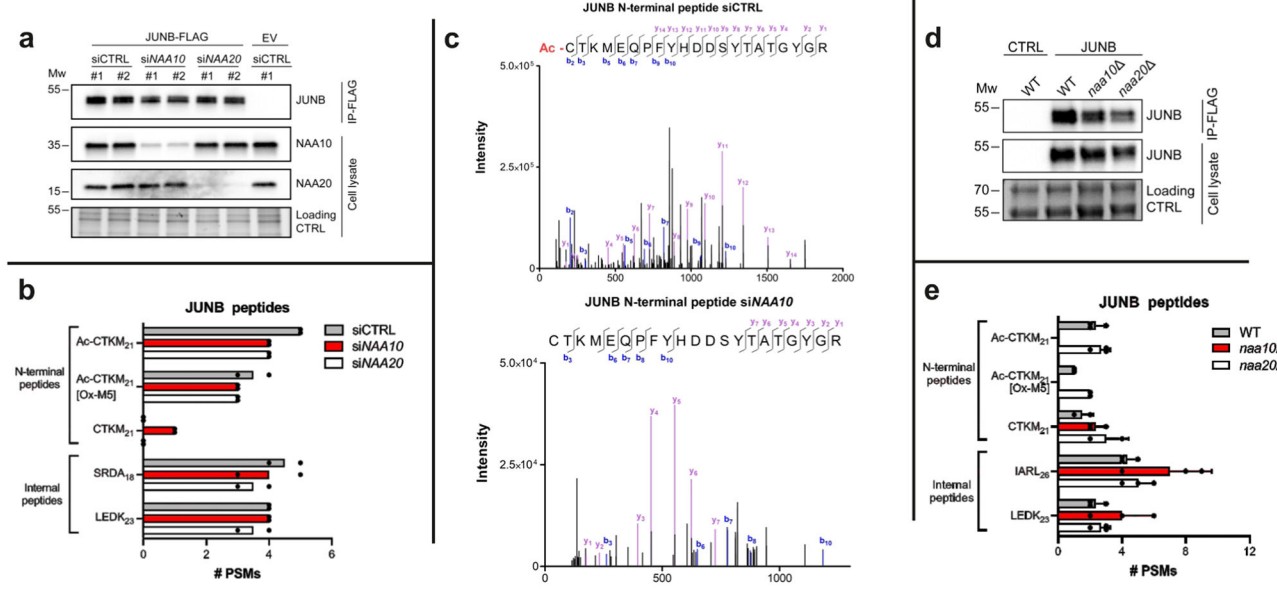

**Fig. 5 | Validation of NatA-catalysed acetylation of Nt-Cys JunB. a** JUNB-FLAG was overexpressed in HAP1 ADO KO cells and treated with either siCTRL, si*NAA10* or si*NAA20*. Cells treated with siCTRL as well as empty vector (EV) were included as controls. Anti-NAA10 and anti-NAA20 were used to verify successful knockdown and disrupted NatA/NatB complexes, whereas anti-JUNB was used to verify successful overexpression and IP. Protein gel image included as loading control. #1 and #2 represent two independent samples. **b** The number of peptide-spectrum matches (PSMs) identified by mass spectrometry for JUNB N-terminal peptides as well as two selected JUNB internal peptides. JUNB peptides designated XXXX$_n$, where XXX are the first four amino acids of the peptide and n is the length of the peptide fragment. Data represent 2 independent biological samples; black dots represent each independent replicate. **c** Example MS/MS spectra for the N-terminal peptide of JUNB in its Nt-acetylated state from siCTRL treated cells and unacetylated state

from si*NAA10* treated cells showing b$_x$ (blue) and y$_x$ (purple) +1 ions with their respective mass/charge ratio and intensity. Free thiols in protein samples were alkylated prior to digestion and LC/MSMS, so Nt-Cys bears a carbamidomethyl group adding 57 Da to the expected mass of b fragment ions. For expanded view see Supplementary Fig. 6. **d** *S. cerevisiae* strains WT, *naa10Δ* and *naa20Δ*, with the last two lacking functional NatA and NatB, respectively, were transformed with a JUNB-FLAG plasmid or control plasmid. Protein gel image included as loading control. **e** The number of PSMs identified by mass spectrometry for JUNB N-terminal peptides as well as two selected JUNB internal peptides. JUNB peptides designated XXXXn, where XXXX are the first four amino acids of the peptide and n is the length of the peptide fragment. Data represent 3 independent biological samples ± standard deviation; black dots represent each independent replicate. MW units = kDa. Source data are provided as a Source Data file.

was seen to increase from 1.2% to 6.0%, in line with NatA preference for acidic residues following the Nt-Cys.

To ascertain whether this NatA substrate specificity is observed in cells, we next compared the in vitro data (Fig. 6a, b) with mass spectrometry Nt-acetylome data from different human cell lines (Supplementary Table 2), and found good alignment between these data sets. Overall, we defined the steady state Nt-acetylation status of seven human Nt-Cys initiating proteins[28,33,35–39]. Six of these, JUNB, AAAS, protein arginine N-methyltransferase 3 (PRMT3), myosin light polypeptide 6 (MYL6), endonuclease III-like protein 1 (NTHL1), and uncharacterized protein FAM241A (FAM241A) were 100% Nt-acetylated. All of these have (M)C-D/S/T Nt-sequences and were Nt-acetylated by NatA in vitro. In contrast, the seventh Nt-Cys starting protein, CHURC1, harbouring a (M)CGD sequence, was found to be 0% Nt-acetylated in human cells (Supplementary Table 2) and was not Nt-acetylated by NatA in vitro (Fig. 6a).

Comparison of the in vitro activities of ADO and NatA towards Nt-Cys-initiating peptides highlights their different substrate preferences (Fig. 6c, Supplementary Table 3). NatA had low activity towards all 'good' ADO substrate peptides (with > 90% activity relative to RGS4). Two of the best NatA substrates in our in vitro assays, MTPN and JUNB, were poor ADO substrates in vitro (3.72% and 4.92% relative to RGS4, respectively (Fig. 2)) and in cells (Fig. 4). However, ADO was moderately active towards other good NatA substrates (PRMT3, AAAS, dynein light chain 1, cytoplasmic (DYNLL1)) in vitro (24.9%, 19.2%, 11.3% relative to RGS4, respectively (Fig. 2)). Overall, our data confirms that acidic or polar residues are preferred by NatA over basic residues after the Nt-Cys. This is distinct from the substrate preferences of ADO, which prefers basic or aromatic residues over acidic residues at the

N-terminus. These preferences are reinforced by increase in NatA activity towards the RGS4 K3D peptide and the decrease in ADO activity towards this substrate both in vitro and in cells. Interestingly, when siRNA was used to knock down NatA activity in cells used for the RGS4 K3D DFOR assay, no impact on RGS4 K3D stability was seen in a cycloheximide chase experiment (Supplementary Fig. 8), suggesting reduced ADO activity towards this substrate was more influential on its stability than increased acetylation. However, it is already known that full knock down of NatA acetylation activity is challenging in human cells (Fig. 5) so it is not possible to be definitive.

## Nt-acetylation abrogates oxidation by ADO in vitro and vice versa

Despite the distinct substrate preferences observed for ADO and NatA, in vitro data indicated that some sequences, such as DYNLL1, PRMT3, and AAAS, have the potential to be modified by either enzyme (Figs. 2, 6). AAAS showed no evidence of ADO-dependent stability in the cellular DFOR assay (Fig. 4), however, we decided to investigate biochemically whether both modifications could occur concurrently on other substrates. NatA-catalysed acetylation takes place on the N-terminal amino group of the peptide[24] whereas ADO-catalysed oxidation takes place on the Nt-Cys thiol sulfur[13]. The modifications are therefore not necessarily mutually exclusive, however as they are both enzyme-catalysed, the presence of either may prevent binding of the other enzyme in order to confer the second modification.

N-terminally acetylated versions of ADO substrate peptides RGS4$_{2-15}$, IL32$_{2-15}$, ANKRD29$_{2-15}$, and RPL4$_{2-15}$ were synthesised; these were some of the peptides towards which ADO had the highest activity in vitro (Fig. 2a). These Nt-acetylated peptides were incubated with

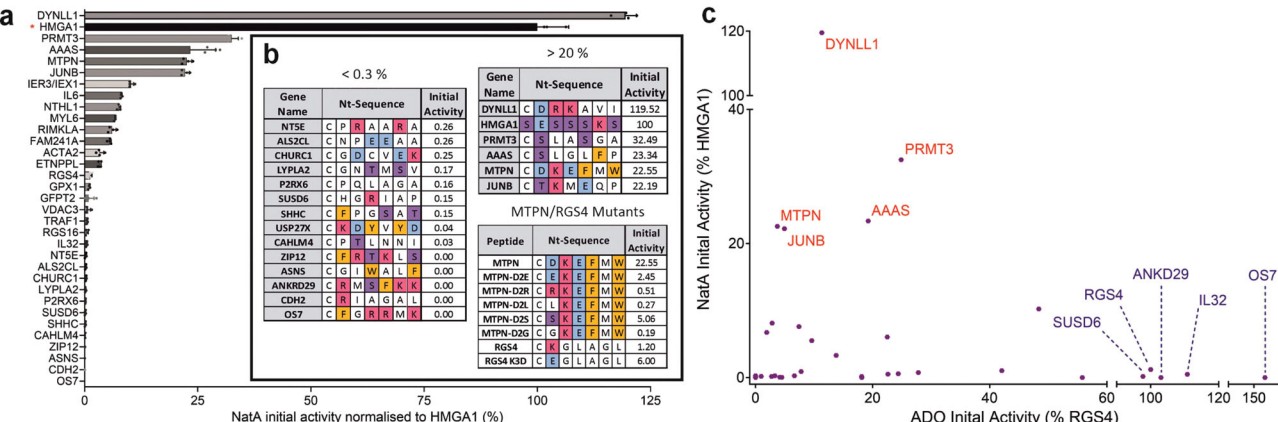

**Fig. 6 | Investigation of NatA-catalysed acetylation of Nt-Cys sequences. a** [$^{14}$C]-Acetyl incorporation into **CX$_6$**RWGRPVGRRRRPVRVYP peptides (where **CX$_6$** is the 1st 7 amino acids of the Met-excised N-terminal sequence of protein of interest) after 5 min (200 μM peptide, 0.3 μM NatA, 200 μM Ac-CoA, 37 °C); activity is normalised to [$^{14}$C]Acetyl incorporation into **SESSSKS**RWGRPVGRRRRPVRVYP (HMGA1) positive control peptide in the same assay (indicated with *). Data represent mean ± standard deviation; $n = 4$. **b** N-terminal sequences of peptides with which NatA was observed to have < 0.3% or > 20% activity compared to HMGA1 positive control peptide. MTPN peptide mutants varying the identity of the amino acid following the Nt-Cys were designed and tested (as in **a**) as well as RGS4 K3D. Amino acids are colour coded according to chemical properties: basic (H/K/R) = red; acidic (D/E) = blue; aromatic (F/W) = orange; polar (S/T) = purple. **c** Activity of NatA (**a**) plotted against activity of ADO (Fig. 2a) for Nt-Cys proteins of interest. Names of best NatA substrate peptides shown in red and names of best ADO substrate peptides shown in blue. Source data are provided as a Source Data file.

recombinant ADO for 30 min at 37 °C. Reactions were quenched with formic acid and peptide modification measured by mass spectrometry. Acetylated RGS4$_{2-15}$ peptide was 1.5% oxidised after 30 min, whereas RGS4$_{2-15}$ peptide with a free N-terminal amine was 99% oxidised in the same time frame (Fig. 7a, b). A similar reduction in oxidation was observed with acetylated IL32$_{2-15}$, ANKRD29$_{2-15}$, and RPL4$_{2-15}$ peptides (Fig. 7b). Therefore, N-terminal acetylation of an ADO substrate leads to an almost complete loss of ADO activity in vitro.

For the converse experiment, oxidised versions of NatA substrate peptides were produced enzymatically using recombinant ADO. Oxidised and non-oxidised 14-mer peptides of JUNB, MTPN and DYNLL1 N-termini were incubated with NatA protein for 90 min. Reactions were quenched with formic acid and analysed by mass spectrometry. JUNB$_{2-15}$ peptide with an N-terminal cysteine sulfinic acid moiety was 2.3% acetylated after 90 min with NatA, whereas JUNB$_{2-15}$ peptide with unmodified Nt-cysteine was 30% acetylated (Fig. 7c, d). A similar reduction in acetylation was also observed for oxidised MPTN$_{2-15}$ and DYNLL1$_{2-15}$ peptides (Fig. 7d). This suggests that N-terminal cysteine oxidation of a substrate greatly reduces NatA activity.

Overall, these data suggest that the two modifications, Nt-Cys thiol oxidation and amino acetylation, are mutually exclusive. Most N-terminal acetylation occurs co-translationally, although some more recently discovered NATs, such as NatF, may act post-translationally[24,37,40]. No enzymes have been identified so far that are able to reverse Nt-Cys acetylation or oxidation. We propose that the most likely application of the observation above is that acetylation could be able to shield some proteins from oxidation by ADO and subsequent degradation via the N-degron pathway. A candidate for this would be AAAS, which was identified as a NatA substrate (Fig. 6), and whose N-terminal sequence did not appear to be an ADO substrate in the DFOR assay (Fig. 4) despite the relevant peptide having comparable ADO activity in vitro to other sequences that were regulated in the DFOR assay.

**Regulation of the stability of endogenous MC-proteins in vivo**
So far, this study has primarily considered the Nt-sequence of candidate ADO substrate proteins up to the 15th residue, both in vitro and in cells. We next considered to what extent these findings predicted ADO-dependent regulation of the intact protein. To this end, wild-type (WT), ADO knock-out (KO) and ADO over-expressing (OE) cell lines

(SH-SY5Y (Fig. 8a)) were produced and analysed for the regulation of those endogenous ADO substrates for which antibodies were available. We used RGS4 and RGS5 as positive controls in these experiments, and immunoblots showed that their stability was both ADO and O$_2$-dependent in SH-SY5Y cells. In contrast, AAAS, ASNS, JUNB, LYPLA2, GPX1 and MTPN levels did not vary substantially between WT, ADO KO and ADO OE cell lines, indicating that these proteins are not endogenously regulated by ADO. This data correlated with previous observations in peptide activity (Fig. 2) and DFOR assays (Fig. 4). The stability of AAAS and JUNB was investigated in WT and ADO knockout HAP1 cells, subjected to si*NAA10*-mediated NatA knockdown (Supplementary Fig. 8). No significant impact on their stability was observed, which may reflect that they are poor ADO substrates (particularly in cells), but likely also reflects residual acetylation activity in these cells (Fig. 4) which may have a protective effect, at least for AAAS as proposed above.

GFPT2, whose N-terminal sequence had low ADO activity in peptide activity assays (Fig. 2) and was not significantly regulated in DFOR assays (Fig. 4b), was not detectable in SH-SY5Y cells. Therefore, we generated similar cell lines in U87-MG cells, in which GFPT2 is highly expressed (Fig. 8b). No difference in protein expression was observed following hypoxic exposure or in ADO KO cells. However, over-expression of ADO did lead to loss of detectable GFPT2, which was partially reversed by hypoxic exposure. These data suggest that endogenous GFPT2 can be regulated by ADO in cells, but only if ADO is present at higher than native levels, which is supported by data from peptide activity assays (Fig. 2). This indicates that other substrates identified in the peptide assay could represent proteins with the potential to be regulated by ADO under certain biological conditions; further investigation of this panel could identify more hypoxically-regulated ADO substrates.

Endogenous proteins for the remaining human sequences studied in DFOR assays, ANKRD29 and SUSD6, could not be detected in any cell lines tested using currently available antibodies. Therefore, cDNA encoding for these proteins was transiently co-transfected in HEK 293T cells, alongside either ADO or CDO1, used here as a negative control[13]. No O$_2$-dependent changes in the stability of transfected ANKRD29 or SUSD6 protein were observed (Fig. 8c). This was unexpected as in vitro and DFOR assay results suggested that ANKRD29 and SUSD6 were among the best ADO substrates (Figs. 2, 4). As transient transfection

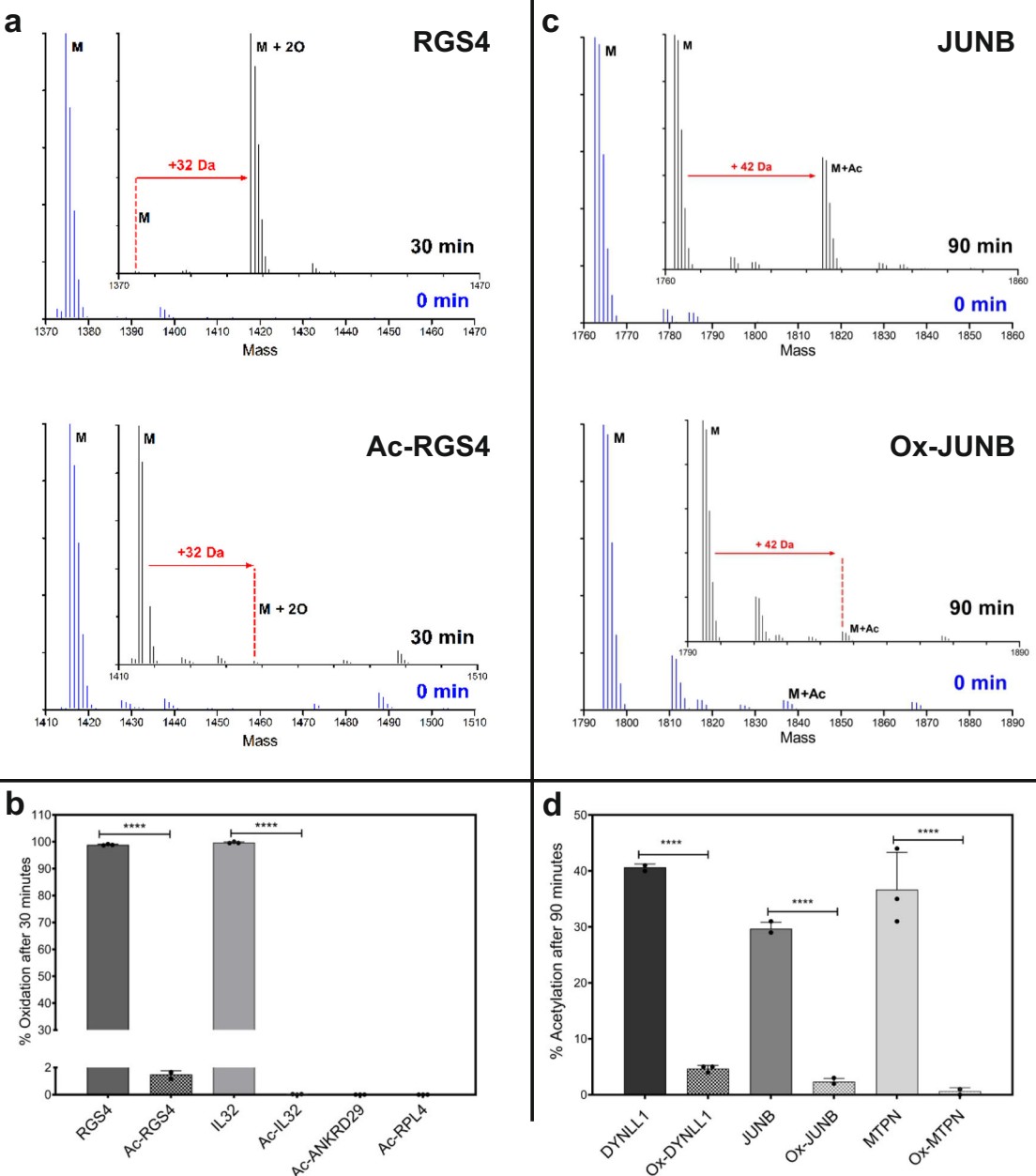

**Fig. 7 | Nt-acetylation abrogates oxidation by ADO in vitro and vice versa.**
**a** Deconvoluted MS spectra showing oxidation (32 Da increase) of RGS4$_{2\text{-}15}$ (RGS4)
and N-terminally acetylated RGS4$_{2\text{-}15}$ (Ac-RGS4) before (blue, 0 min) and after
(black, 30 min) incubation with HsADO at 37 °C (100 μM peptide, 0.1 μM ADO).
**b** Oxidation of other acetylated ADO substrate peptides (Fig. 2a) compared to
oxidation of RGS4$_{2\text{-}15}$ and IL32$_{2\text{-}15}$ peptides after 30 min (100 μM peptide, 0.1 μM
HsADO, 37 °C). **c** Deconvoluted MS spectra showing acetylation (42 Da increase) of
JUNB$_{2\text{-}15}$ (JUNB) and N-terminally oxidised JUNB$_{2\text{-}15}$ (Ox-JUNB) before (blue, 0 min)

and after (Ox, 30 min) incubation with NatA at 37 °C (100 μM peptide, 0.15 μM
NatA). **d** Acetylation of other oxidised NatA substrates peptides (Fig. 6a) compared
to acetylation of non-oxidised NatA substrates peptides after 90 min (100 μM
peptide, 0.15 μM NatA, 37 °C). In (**b, d**), data are presented as mean ± standard
deviation, $n = 3$. Statistical significance determined using 2-way ANOVA with Bon-
ferroni's multiple comparison post-test: ****$p \leq 0.0001$. Source data are provided as
a Source Data file.

typically leads to very high expression of target proteins, which may
result in dysregulation via endogenous pathways, SH-SY5Y cell lines
stably expressing either ANKRD29:FLAG or SUSD6:FLAG were gener-
ated. Much like the assays in HEK293T cells, levels of ANKRD29 and
SUSD6 also did not vary detectably in an ADO-dependent manner in
SH-SY5Y cells (Fig. 8d, e). It is possible that these proteins were not
destabilised due to inaccessibility of their N-termini for ADO-catalysed
oxidation: In rice, the ERF-VII SUB1A-1 is not regulated by the N-degron
pathway despite having an Nt-Cys[9], likely due to interaction between
its N- and C-termini which result in the N-terminus being unavailable
for modification[41]. Similar intra- (protein folding) or intermolecular

(protein-protein) interactions may be occurring in full-length
ANKRD29 and SUSD6 proteins, preventing their destabilisation by
Cys/Arg N-degron components in vivo, including ADO, due to hin-
dered access to their N-terminal residues. Confined subcellular loca-
lisation relative to ADO may also explain why full-length versions of
ANKRD29 and SUSD6 were not proteolytically regulated, and further
work will be required to resolve this.

In this study, we have shown that ADO is highly selective for
certain Nt-Cys sequences in vitro, preferring basic or aromatic residues
after the Nt-Cys (Fig. 2 and Fig. 3). Cellular assays using a DFOR system
broadly supported observations from in vitro ADO assays and

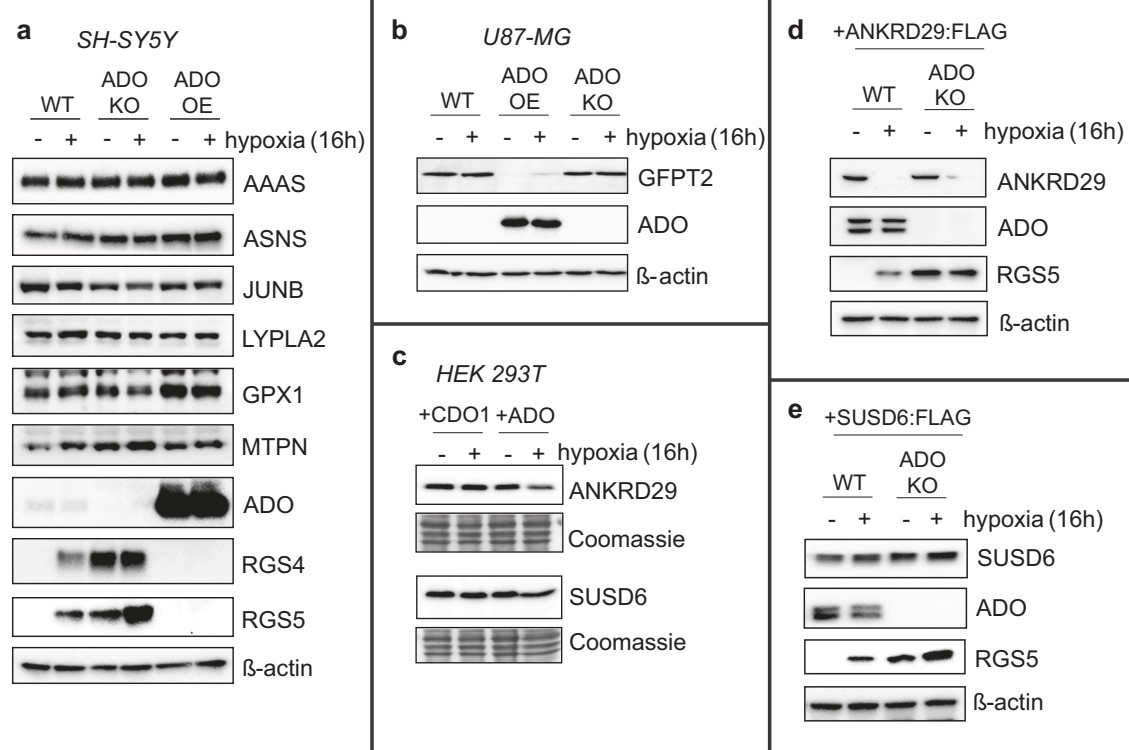

**Fig. 8 | Regulation of putative ADO substrates human cells. a, b** WT, ADO KO or ADO overexpressing (OE) SH-SY5Y (**a**) or U87-MG (**b**) were exposed to hypoxia (1% O₂) for 16 h, then protein samples were collected and analysed for the expression of various potential ADO substrates. RGS4 and RGS5 were used as positive controls. **c** HEK 293T cells were transiently co-transfected with cDNAs encoding ANKRD29 or SUSD6, alongside either CDO1 or ADO, and subjected to hypoxia as before. **d, e** SH-SY5Y cells stably expressing either ANKRD29:FLAG (**d**) or SUSD6:FLAG (**e**) were

subjected to hypoxia for 16 h and the expression of both proteins was analysed using an anti-FLAG antibody. RGS5 expression was also analysed as a positive control. SUSD6 was detected at a much higher level relative to RGS5 and ANKRD29 in a stably polyclonal population. All blots (**a–e**) are representative of at least 3 independent experiments. Source data (including relevant molecular weight markers) are provided as a Source Data file.

confirmed that ADO is also selective among Nt-Cys sequences *in cellulo* (Fig. 4). While investigating discrepancies between in vitro and *in cellulo* data, we demonstrated that NatA is responsible for Nt-Cys acetylation, and is also selective amongst Nt-Cys sequences in vitro. Interestingly, NatA's substrate preferences were distinct from those of ADO, favouring instead acidic or polar residues immediately after the Nt-Cys (Fig. 5). Substrates of NatA identified in vitro included AAAS, JUNB and MTPN. These sequences all stabilised GFP in an ADO-independent manner in DFOR assays, suggesting that Nt-Cys acetylation could be a stabilising modification. We also showed, in vitro, that Nt-Cys acetylation can block oxidation by ADO and that Nt-Cys oxidation can block acetylation by NatA (Fig. 6). While the possibility of dual regulation may be conditionally relevant to the stability of some proteins, Nt-acetylation by NatA is currently thought to be co-translational and irreversible[24]. Therefore, although it is possible that NatA could shield potential ADO substrates from oxidation and subsequent degradation, the divergent substrate repertoires of NatA and ADO mean that this is unlikely to be a common phenomenon. Cellular assays with full-length putative ADO substrate proteins confirmed that ADO is also selective amongst Nt-Cys sequences in vivo, but some results differed from in vitro and *in cellulo* data, suggesting other factors (such as tertiary protein structure) may be influencing ADO activity (Fig. 7). It is also worth noting that our cellular protein stability assays were only capable of identifying ADO substrates that were also substrates of the remaining components of the Arg/N-degron pathway, Arg-tRNA transferase 1 and E3 ubiquitin ligases UBR1/2[42]. Therefore, it is formally possible that all in vitro peptide substrates could also be ADO substrates in vivo. If this were the case it could imply an alternative role for ADO-catalysed Nt-Cys oxidation. Exploring this was

beyond the scope of this study but could be an interesting future avenue of investigation.

Overall, this study shows that ADO and NatA prefer distinct Nt-Cys substrates based on sequence characteristics. This can contribute to, but not define, oxygen-dependent protein stability in vivo. Other factors, such as overall substrate protein structure and other N-terminal modifications, should be considered when determining the biological relevance of in vitro observations. Although sequence determinants have been identified using short fragments of protein N-termini, which may (as demonstrated) not represent outcomes with full-length proteins, protein N-termini are often exposed in full-length proteins[43] and upon emergence from the ribosome (at the point of co-translational Nt-acetylation[24]). Their regulation by stabilising or destabilising post-translational modifications is therefore an important determinant of protein function. This study serves to demonstrate the complex interplay that can exist between N-terminal Cys oxidation and acetylation, and the implications this can have for protein stability and downstream hypoxic signalling.

## Methods
### ADO in vitro assays
**Expression and purification.** Recombinant His-tagged *Hs*ADO was produced as described previously[13]. Briefly, HsADO was overexpressed in Rosetta (DE3) *E.coli* cells (Merck) then purified using Ni(II)-affinity chromatography. Following imidazole removal using a desalting column, recombinant *Hs*ADO protein was further purified by size exclusion chromatography prior to storage of protein aliquots at −80 °C. Protein purity was verified by SDS-PAGE and protein concentration estimated using A₂₈₀ₙₘ measurements.

**Peptide synthesis and purification.** Peptides were purchased from GL Biochem or Genscript, with the exception of nine peptides (Supplementary Table 4), which were synthesised using a Liberty Blue™ Microwave Peptide Synthesiser (CEM Corporation), via fluorenylmethyloxycarbonyl (Fmoc) solid phase peptide synthesis using a NovaPEG rink amide resin (Merck). The 14-mer peptides were prepared as C-terminal amides and cleaved from the resin using trifluoroacetic acid/water/triisopropylsilane/dimethoxy benzene mix. Peptides were purified using high-performance liquid chromatography and a C18 column. Peptide purity was assessed by matrix-assisted laser desorption/ionisation-time of flight mass spectrometry (MALDI-TOF MS, Supplementary Fig. 9). Peptide samples were prepared by dissolution in a mix of acetonitrile and water (using the minimum percentage of acetonitrile at which peptide remained soluble). The peptide sample (0.8 μL) was spotted with α-cyano-4-hydroxycinnamic acid (CHCA MALDI matrix) (0.8 μL) onto an MSP 96 target polished steel BC plate (Bruker) and analysed using an Autoflex™ Speed MALDI-TOF mass spectrometer (Bruker) operated in positive reflectron mode.

Active peptides were quantified by NMR (Supplementary Fig. 10). NMR samples were prepared by adding 5 μL peptide (10 or 50 mM in water) and 5 μL 3-(trimethylsilyl)propionic-2,2,3,3-$d_4$ acid (TSP) (58 mM in $D_2O$) to 500 μL $D_2O$, then analysed using $^1$H NMR spectroscopy on the AVIII HD 500 (Bruker) with 1D-$^1$H-nuclear Overhauser effect spectroscopy (1D-NOESY) solvent suppression. NMR signals for methyl groups on aliphatic peptide residues were used to determine the concentration of the peptide relative to the TSP standard ($^1$H chemical shift = 0 ppm). Spectra were generated in TopSpin 3.2 and integrated manually in MestReNova v14.1.

N-terminally oxidised peptides were produced by incubating JUNB, MTPN or DYNLL1 peptide (1 mM) with recombinant His-tagged *Hs*ADO (1 μM (JUNB, DYNLL1) or 10 μM (MTPN)) in assay buffer (50 mM BisTris Propane, 50 mM NaCl, pH 7.5) supplemented with cofactors (5 mM TCEP, 20 μM $FeSO_4$, 1 mM sodium ascorbate) for 1 h at 37 °C using a bench top thermomixer. Oxidised peptide was extracted using a centrifugal filter (Amicon, 10 kDa MWCO, 0.5 mL) and stored at -20 °C. Oxidation was confirmed by high-throughput MS using a RapidFire RF360 sampling robot (Agilent) connected to an Agilent 6530 Accurate-Mass Q-ToF mass spectrometer operated in positive ion mode. Source conditions were adjusted to maximise sensitivity and minimise fragmentation. Samples were injected onto a C-4 solid phase extraction (SPE) cartridge equilibrated with deionised water containing 0.1% formic acid (v/v), washed with the equilibration solution and eluted in 85% acetonitrile and 0.1% formic acid (v/v). Spectra were assessed in MassHunter Qualitative Analysis B.07.00 (Agilent).

**Activity assays.** Substrate peptides (100 μM) were incubated with recombinant His-tagged *Hs*ADO (0.1 μM) in assay buffer (50 mM Bis-Tris Propane, 50 mM NaCl, pH 7.5) supplemented with cofactors (5 mM TCEP, 20 μM $FeSO_4$, 1 mM sodium ascorbate) at 37 °C using a bench top thermomixer. At required time points 5 μL of reaction mixture was added to 45 μL 1% formic acid (v/v) to quench the reaction. Oxidation was monitored by high-throughput MS as described above. Turnover was quantified by comparing the integrated area underneath the product and substrate ions extracted from the total ion current chromatogram using the RapidFire Integrator software (Agilent). Spectra were manually assessed in MassHunter Qualitative Analysis B.07.00 (Agilent) to ensure the correction was chosen for quantification.

Product and substrate ions for OS1,2, 6-10 peptides could not be detected by high-throughput MS as described above, so oxidation of these peptides was monitored using an Agilent 1290 Infinity II LC System connected to an Agilent 6530 Accurate-Mass Q-ToF mass spectrometer operated in positive ion mode. Samples were injected onto a ProSwift™ RP-4H (Thermo Scientific™) and eluted in a gradient of 5 to 100% acetonitrile (0.1% formic acid (v/v)) over 5 min. Turnover

was quantified using the integrated area underneath the product and substrate ions extracted from the total ion current chromatogram using Agilent Qualitative Analysis software (B.07.00).

## Docking studies with Nt-Cys peptides

Pymol was used to add 7-mer Nt-Cys peptides to the structure of *Mm*ADO ((7LVZ, chain D [https://www.rcsb.org/structure/7LVZ] (crystal structure of *Mm*ADO)), bound to the active site Fe via the Nt-Cys thiol (as observed in spectroscopic experiments[44]). Map files were generated using AutoGrid. AutoDock 4.2[45] experiments were conducted setting 7LVZ_D (minus Fe) as the 'Rigid' macromolecule and Fe bound to 7-mer peptide as the 'Flexible Residues'. A water molecule was used as the ligand. For each complex, 100 docking experiments were performed using the Lamarckian genetic algorithm with the default parameters. A maximum of 2.5 million energy evaluations was applied for each experiment.

## ADO in vivo experiments

Cellular experiments were conducted in SH-SY5Y and HEK293T cells (sourced as described previously[13]), and in U87-MG cells obtained from the Francis Crick Institute core cell service. Cells were cultured in either DMEM-F12 (SH-SY5Y) or DMEM, supplemented with 10% foetal bovine serum, 2 mM L-Glutamine, 100 U/ml penicillin and 10 μg/ml streptomycin. Cells were maintained at 37 °C under an atmosphere of 5% $CO_2$ in air. Hypoxic incubations were conducted within an atmosphere-regulated workstation set at 1% $O_2$: 5% $CO_2$: balance $N_2$ (*Invivo* 400, Baker-Ruskinn Technologies).

**Dual fluorescence oxygen reporter assays.** $MCX_{13}$-GFP:P2A:mCherry constructs were generated for $MCX_{13}$ = *Hs*RGS4$_{1-15}$; *Hs*ANKRD29$_{1-15}$; *Hs*SUSD6$_{1-15}$; *Hs*GPX1$_{1-15}$; *Hs*AAAS$_{1-15}$; *Hs*GFPT2$_{1-15}$; *Hs*JUNB$_{1-15}$; *Hs*MTPN$_{1-15}$; *Hs*LYPLA2$_{1-15}$; *Hs*ASNS$_{1-15}$; OS7 and *At*RAP2.12$_{1-15}$. To generate these plasmids a GFP:P2A:mCherry plasmid template was first produced by ligation of a synthetic oligonucleotide into pcDNA3.1 (GeneArt, Thermo Fisher Scientific).

$MCX_{13}$-GFP constructs were generated by PCR and restriction enzyme cloning, using the primers listed in Supplementary Table 5 and a pcDNA3.1-GFP:P2A:mCherry template plasmid[27]. 2 μg plasmid was transfected into SH-SY5Y cells using GeneJuice (Novagen) or Lipofectamine™ 3000 (Invitrogen™) according to manufacturer's protocol. After 24 h cells were treated with 0.5 mg/mL G418 antibiotic (Santa Cruz Biotechnology) for 2-4 weeks to generate a polyclonal population. Successful expression was verified by mCherry fluorescence. Reporter stability was assayed as reported recently[27]. In brief, cells were seeded in a 96-well plate and allowed to reach 80% confluency. Medium was replaced with DMEM without phenol red (supplemented with 5% fetal bovine serum, 2mM L-glutamine, 100 U/ml penicillin and 10 μg/ml streptomycin), containing either 2,2-dipyridyl (200 μM) or DMSO (vehicle control) and incubated in normoxia or hypoxia for 20-24 h. After incubation, GFP and mCherry fluorescence were measured by multichromatic fluorescence imaging using a FLUOstar (BMG Lab-Tech); gain was adjusted to prevent saturation of mCherry signal and kept consistent across biological repeats. The GFP fluorescence was measured at 485 nm excitation, 520 nm emission and mCherry at 544 nm excitation, 590 nm emission. The GFP:mCherry ratio was calculated using blank-adjusted values. Although the expected GFP:mCherry ratio would be 1 in the absence of ADO-mediated destabilisation, differences in the intrinsic fluorescence characteristics between GFP and mCherry meant that this was not the case in practice. Moreover, due to the considerable range of fluorescence measured between constructs, experiments were performed in batches with different gain parameters. To account for this variability between assays, results for each experiment were normalised to the GFP:mCherry ratio obtained in cells expressing the positive control construct RGS4$_{(1-15)}$DFOR.

**Assay of full-length protein stability in cells.** The stability of candidate ADO substrates in their full-length form was assessed in parental SH-SY5Y (ECACC 94030304, human female) or U87-MG cells (ATCC CRL-3216, human female) and in those in which ADO was genetically silencing using CRISPR-Cas9[13]. ADO overexpressing cells were generated by infecting ADO KO cells with pRRL-FLAG:ADO-IRES-GFP lentivirus, producing a polyclonal population with > 90% GFP positivity after 1 week. WT, ADO KO or ADO overexpressing cells were seeded in 6-well plates and allowed to reach 80% confluency before exposure to 1% $O_2$ (as above) for 16 h. To assess the stability of full length of SUSD6 and ANKRD29, which were not detectable in any cell lines examined, HEK293T cells were transiently transfected with 1 µg pcDNA3.1-SUSD6:FLAG or -ANKRD29:FLAG plasmids (Genscript, USA) alongside pRRL-FLAG-ADO or -CDO1 (as a control) using polyethyleneimine. 24 h later, cells were split into 2×6-well plates and allowed to adhere before exposure to 1% $O_2$ for 16 h. In parallel, WT or ADO KO SH-SY5Y cells were transfected with pcDNA3.1-SUSD6:FLAG or -ANKRD29:FLAG plasmids and polyclonal stably expressing populations generated as before. Protein samples were collected using Igepal lysis buffer (10 mM Tris pH 7.5, 0.25 M NaCl, 0.5% Igepal) containing Complete™ protease inhibitor cocktail (Sigma Aldrich) at 4 °C for 5 min. Samples were processed and immunoblotted as described elsewhere[13] using the following antibodies; AAAS (1:1000 dilution, sc-374073), ASNS (1:1000 dilution, sc-365809), LYLPLA2 (1:1000 dilution, sc-390546), MTPN (1:500 dilution, sc-166072), RGS5 (1:500 dilution, sc-514184), and horse radish peroxidase-conjugated FLAG (1:1000 dilution, M2 A8592) from Santa Cruz Biotechnology, JunB (1:1000 dilution, 3753), GPx1 (1:500 dilution, 3286) and RGS4 (1:1000 dilution, 15129) from Cell Signalling Technologies, GFPT2 (1:1000 dilution, NBP1-56688) and SUSD6 (1:1000 dilution, H00009766-B01P) from Novus Biotech, ANKRD29 (1:1000 dilution, 23999-1-AP) from Proteintech, and ADO (1:1000 dilution, ab13243) horse radish peroxidase-conjugated β-actin (1:25,000 dilution, ab49900) from Abcam. HRP-conjugated secondary antibodies (1:1000 dilution) were supplied by DAKO.

## NatA in vitro experiments
**Expression and purification.** The pFastBac Dual plasmid for co-expression of human NAA10 residues 1-160 and 6xHis-tagged full-length NAA15 was a gift from R. Marmorstein lab. This NatA construct was expressed and purified essentially as described by Gottlieb and Marmorstein[46]. Briefly, the NatA coding sequence was transferred into EMBacY bacmid in *E. coli*[47], the purified bacmid was transfected into Sf9 cells to generate the baculovirus, and the virus was used to infect fresh Sf9 cells to express NatA. NatA was purified in three steps with $Ni^{2+}$-affinity, ion exchange, and size exclusion chromatography. Purified protein was quantified based on $A_{280nm}$, flash frozen in liquid nitrogen and stored at -80 °C until use.

**Activity assays (screening).** The peptides for the NatA assays with seven residues from the target substrate followed by 17 standard residues (RWGRPVGRRRRPVRVYP) were purchased from Innovagen. The NatA activity assays were adapted from Drazic & Arnesen[34]. Using the referenced assay with Cys-containing peptides resulted in high background that masked any enzyme-generated signal, likely caused by chemical acetylation[48]. To overcome this, we shortened the reaction time to 5 min, increased the enzyme concentration to 300 nM, and changed the reaction buffer to one with 50 mM Na-phosphate pH 7.0, 100 mM NaCl, 1 mM EDTA, 1 mM DTT and 2 mg/ml BSA. Otherwise, the assays were performed as described. Briefly, Ac-CoA concentration, 1/4 of which was [acetyl-1-$^{14}$C] labelled (Perkin Elmer for labelled, Sigma for unlabelled), and peptide concentration were both 200 µM. 20 µL reactions were incubated at 37 °C and stopped by transfer to P81 filter paper. Papers were washed three times with 10 mM HEPES pH 7.4, dried, moved to scintillation vial with Ultima Gold F scintillation solution (Perkin Elmer) and measured with Tri-Carb 3100TR

scintillation counter (Packard). DPM values of two control reactions that received enzyme buffer instead of enzyme were subtracted from the average DPM of the four enzyme reactions and the activity was presented as the percentage of the activity towards the HMGA1 control peptide.

**Activity assays with oxidised substrates.** Oxidised and non-oxidised $JUNB_{2-15}$, $MTPN_{2-15}$ and $DYNLL1_{2-15}$ peptides (100 µM) were incubated with recombinant NatA (0.15 µM) and acetyl coenzyme A (100 µM) in assay buffer (50 mM Na-phosphate, 100 mM NaCl, pH 7.0) supplemented with TCEP (1 mM) at 37 °C using a bench top thermomixer. At required time points 5 µL of reaction mixture was added to 45 µL 1% formic acid (v/v) to quench the reaction. Acetylation was monitored by high-throughput MS as described above and turnover quantified using RapidFire Integrator software as described above. Background acetylation was measured using no enzyme controls and extracted from integrated data before calculating turnover.

## NatA in vivo experiments
**LC-MS/MS analysis of immunoprecipitated JUNB-FLAG.** HAP1 ADO KO cells (Horizon Discovery, HZGHC00812c004) were seeded with a cell density of 3 million cells per 10 cm dish in IMDM medium. After 6-7 h incubation at 37 °C and 5% $CO_2$, cells were transfected with 25 nM siNAA10, siNAA20 or non-targeting control siRNA (siCtrl) using DharmaFECT 1 Transfection reagent (Horizon Discovery) according to manufacturer´s protocol. After 48 h, the cells were split in a 1:2 ratio and incubated for 6-7 h before transfecting the cells with 6 µg pRP/JUNB-FLAG-P2A-EGFP plasmid using XtremeGENE 9 DNA Transfection Reagent according to manufacturer´s protocol. As a control, cells were transfected with empty vector (EV). 24 h after plasmid transfection, cells were washed twice in ice cold 1 × PBS before harvested by cell scraping in 0.5 ml 1 x PBS. For each sample, two 10 cm dishes with uniform conditions were combined. Cells were centrifuged at 4 °C and 17,000 × g for 15 sec to collect cell pellets. The cell pellets were resuspended in 200 µl of lysis buffer (10 mM Tris/Cl pH 7.5, 150 mM NaCl, 0.5 mM EDTA and 0.5% NP-40, 1x complete EDTA-free protease inhibitor cocktail). The cell debris was pelleted by centrifugation at 4 °C and 17,000 × g for 5 min. The supernatant was collected, and JUNB-FLAG was immunoprecipitated using 30 µL of ChromoTek DYKDDDDK Flag-Trap Agarose beads (Proteintech) according to the manufacturer´s protocol. Samples were analysed by Western blotting to verify successful siRNA knockdown, plasmid transfection and IP. The following antibodies were used: NAA10 (1:2000 dilution, Cell Signaling, 13357), NAA20 (1:1000 dilution, Sigma Aldrich, HPA063344) and JUNB (1:1000 dilution, Abcam, ab128878).

The IP samples were further processed for LC-MS/MS analysis. IP beads were washed twice in cold 1 x PBS prior to on-bead digestion. IP beads were resuspended in 30 µL of incubation buffer (50 mM Tris/Cl pH 7.6, 5 mM $CaCl_2$ and 2 mM EDTA) supplemented with 10 mM dithiothreitol (DTT) for 10 min at 95 °C. After the beads were cooled down, cysteines were alkylated by incubation with 30 mM chloroacetamide (CAA) for 1 h at 1000 rpm shaking. Afterwards, 15 mM of DTT was added to quench the remaining CAA and to activate the protease Arg-C (Promega, V1881), which was added in a 1:100 ratio (µg protease/µg protein). Protein samples were digested overnight at 37 °C on a shaker at 1000 rpm. The next day, the peptides were separated from the beads, which included a wash with the incubation buffer to collect potential lost peptides. The digestion reaction was stopped by adding trifluoroacetic acid (TFA) to a final concentration of 1%. Peptides were subsequently desalted using Pierce C18 spin tips (Thermo Scientific) and eluted in 100 µL of 0.1% formic acid (FA) and 70% acetonitrile (ACN). The samples were further frozen at −80 °C, before drying the peptides using a speed vac evaporator. Finally, the dried

peptides were resuspended in 10 μL 2% ACN and 0.5% FA and sonicated for 30 s to enhance the solubilization of the peptides. The peptide samples were analysed by LC-MS/MS.

About 0.5 μg peptide sample was injected into an Ultimate 3000 RSLC system (Thermo Scientific, Sunnyvale, California, USA) connected online to a Orbitrap Eclipse mass spectrometer (Thermo Scientific, Bremen, Germany) equipped with EASY-spray nano-electrospray ion source source (Thermo Scientific).

The sample was loaded and desalted on a pre-column (Acclaim PepMap 100, 2 cm × 75 μm ID nanoViper column, packed with 3 μm C18 beads) at a flow rate of 5 μL/min for 5 min with 0.1% trifluoroacetic acid.

Peptides were separated during a biphasic ACN gradient from two nanoflow UPLC pumps (flow rate of 200 nl/min) on a 25 cm analytical column (PepMap RSLC, 25 cm × 75 μm ID. EASY-spray column, packed with 2 μm C18 beads). Solvent A and B were 0.1% FA (vol/vol) in water and 100% ACN, respectively. The gradient composition was 5%B during trapping (5 min) followed by 5-7%B over 1 min, 7–20% B for the next 44 min, 20–32% B over 12 min, and 32–80% B over 3 min. Elution of very hydrophobic peptides and conditioning of the column were performed for 14 min isocratic elution with 80% B and 15 min isocratic conditioning with 5% B, respectively. Instrument control was through Thermo Scientific SII for Xcalibur 1.6.

The FAIMS Pro interface performs gas-phase fractionation, enabling preferred accumulation of multiply charged ions to maximise the efficiency of data-dependent acquisition (DDA) routines and increase proteome coverage. Short-ion residence time in the FAIMS Pro interface electrode assembly enables use of multiple CV settings in a single run to increase proteome coverage.

Peptides eluted from the column were detected in the Orbitrap Eclipse Mass Spectrometer with FAIMS enabled using two compensation voltages (CVs), −50V and −70V, respectively. During each CV, the mass spectrometer was operated in the DDA-mode (data-dependent-acquisition) to automatically switch between one full scan MS and MS/MS acquisition. Instrument control was through Orbitrap Eclipse Tune 3.5 and Xcalibur 4.5. The cycle time was maintained at 1.5 s/CV. MS spectra were acquired in the scan range 375–1500 $m/z$ with resolution $R = 120\,000$ at $m/z$ 200, automatic gain control (AGC) target of 8e5 and a maximum injection time (IT) set to Auto. The most intense eluting peptides with charge states 2 to 6 were sequentially isolated to a target value (AGC) of 5e4 and a maximum IT of 75 ms in the C-trap, and isolation width maintained at 1.6 $m/z$ (quadrupole isolation), before fragmentation in the HCD (Higher-Energy Collision Dissociation. Fragmentation was performed with a normalised collision energy (NCE) of 30%, and fragments were detected in the Orbitrap at a resolution of 15,000 at $m/z$ 200, with first mass fixed at $m/z$ 110. One MS/MS spectrum of a precursor mass was allowed before dynamic exclusion for 30 s with "exclude isotopes" on. Lock-mass internal calibration was not enabled. The spray and ion-source parameters were as follows. Ion spray voltage = 1800V, no sheath and auxiliary gas flow, and capillary temperature = 275 °C. The data obtained from the LC-MS/MS was analysed using Proteome Discoverer.

**LC-MS/MS analysis of immunoprecipitated JUNB-FLAG from yeast strains.** Expression vectors pBEVY-U-JUNB-FLAG and pBEVY-U-EV (serving as a control) were transformed into the three different *S. cerevisiae* strains: MATa strain BY4742 (Y10000, EUROSCARF) serving as WT strain, *NAA10*::kanMX4 (Y10976, EUROSCARF) without functional NatA, and *NAA20*:kanMX4 (Y15546, EUROSCARF) without functional NatB. The yeast strains were selected and grown at 30 °C on plates without uracil. For culturing, the strains were grown in 25 ml synthetic complete medium without uracil (SC-Ura) for 48 h before the culture was diluted to 50 ml to obtain an $OD_{600}$ of 0.9. When reached exponential growth phase ($OD_{600} = 2.5–3$) the cells were harvested by centrifugation at $3095 \times g$ for 15 min at 4 °C and washed twice in 1xPBS. To lyse the cells 500 μL of lysis buffer (50 mm Tris-HCl pH 7.6, 12 mm

EDTA, 250 mm NaCl, 140 mm $Na_2HPO_4$ supplemented with 1x EDTA-free protease inhibitor) and 0.4 g of acid washed 425–600 μm glass beads (Sigma) were added to the cells before vortexing 10 ×30 s with 30 s on ice between each round. The lysates were centrifugated at $7500 \times g$ for 10 min at 4 °C to pellet cell debris, and the supernatant was collected. Immunoprecipitation of JUNB-FLAG, on-bead digestion and LC-MS/MS analysis was performed as described above with the following modifications; Peptide samples were desalted using the Oasis WAX 96-well μElution Plate, 2 mg (Waters) and run on the Exploris 480 mass spectrometer (Thermo Scientific, Bremen, Germany) equipped with EASY-spray nano-electrospray ion source (Thermo Scientific). Peptides were separated during a biphasic ACN gradient from two nanoflow UPLC pumps (flow rate of 250 nl/min) on a 25 cm analytical column (PepMap RSLC, 25 cm × 75 μm ID. EASY-spray column, packed with 2 μm C18 beads). Solvent A and B were 0.1% FA (vol/vol) in water and 100% ACN, respectively. The gradient composition was 5%B during trapping (5 min) followed by 5–8%B over 1 min, 8–28%B for the next 42 min, 28–40%B over 15 min, and 40–85%B over 2 min. Elution of very hydrophobic peptides and conditioning of the column were performed for 8 min isocratic elution with 85%B and 15 min isocratic conditioning with 5%B, respectively. Instrument control was through Thermo Scientific SII for Xcalibur 1.6.

Peptides eluted from the column were detected in the Exploris 480 Mass Spectrometer with FAIMS enabled using two compensation voltages (CVs), -50V and -70V, respectively, and "Advanced Peak Determination" on. During each CV, the mass spectrometer was operated in the DDA-mode (data-dependent-acquisition) to automatically switch between one full scan MS and MS/MS acquisition. Instrument control was through Orbitrap Exploris 480 Tune 3.1 and Xcalibur 4.4. The cycle time was maintained at 1.2 s/CV. MS spectra were acquired in the scan range 375–1500 $m/z$ with resolution $R = 120\,000$ at $m/z$ 200, automatic gain control (AGC) target of 3e6 and a maximum injection time (IT) at auto (depending on transient length in the orbitrap). The most intense eluting peptides with charge states 2 to 5 were sequentially isolated to standard target value (AGC, 1e5) and a maximum IT of 75 ms in the C-trap, and isolation width maintained at 1.6 $m/z$ (quadrupole isolation), before fragmentation in the HCD (Higher-Energy Collision Dissociation). Fragmentation was performed with a normalised collision energy (NCE) of 30%, and fragments were detected in the Orbitrap at a resolution of 15,000 at $m/z$ 200, with first mass fixed at $m/z$ 120. One MS/MS spectrum of a precursor mass was allowed before dynamic exclusion for 30 s with "exclude isotopes" on. Lock-mass internal calibration was not enabled. The data obtained from the LC-MS/MS was analyzed using Proteome Discoverer.

**Cycloheximide chase assay.** 75 000 HAP1 WT cells (Horizon Discovery, C631) and HAP1 ADO KO cells (Horizon Discovery, HZGHC008126c004) were seeded in a 6 well plate and grown overnight at 37 °C in Iscove´s Modified Dulbecco´s Medium (IMDM) containing 10% Fetal Bovine Serum (FBS) and 1% Penicillin-Streptomycin. The next day, cells were transfected with 50 nM si*NAA10* or non-targeting control siRNA (siCTRL) using DharmaFECT 1 Transfection reagent (Horizon Discovery) according to manufacturer´s protocol. After a 72 h incubation, cells were treated with 50 μg/mL cycloheximide (CHX) and harvested at different time-points ranging from 0 h – 8 h. Cells harvested at time point 0 h were not treated with CHX and served as a baseline. The cells were washed twice in ice cold 1 x PBS and harvested by scraping in 1 mL 1 x PBS. Cells were collected by a 15 s centrifugation at 4 °C and $17,000 \times g$. The cell pellets were lysed for 30 min on ice in 200 μL IPH lysis buffer (50 mM Tris-HCl pH 8.0, 150 mM NaCl, 5 mM EDTA, 0.5% NP-40, 1x complete EDTA-free protease inhibitor cocktail). The cell debris was pelleted by a 5 min centrifugation at 4 °C at $17,000 \times g$, and the supernatant was collected. The protein concentration was measured using Pierce BCA protein Assay Kit (Thermo Fisher Scientific) and adjusted to an equal protein

concentration for each cell lysate. The protein turnover of AAAS and JUNB was analysed by Western blotting using the following antibodies; AAAS (1:2000 dilution, Bethyl Laboratories, A304-514A), JUNB (1:1000 dilution, Abcam, ab128878), NAA10 (1:2000 dilution, Cell Signaling, 13357) to verify knockdown, and gel images was used as a loading controls. The secondary antibody used was HRP-linked Rabbit IgG (1:5000 dilution, Cytiva, NA934). The amount of AAAS and JUNB at each time point was quantified relative to the amount present at time point 0 h and normalised to the loading control.

To measure the stability of the RGS4 K3D DFOR construct, SH-SY5Y cells stably expressing pcDNA3- RGS4$_{1-15}$(K3D):DFOR (generated as described above) were seeded at 50000 cells into 35 mm dishes and transfected with siNAAT10 or scr control as described above. 48 h post-transfection, cells were re-seeded into a 96-well plate (6 wells per condition) and allowed to recover for 24 h. At $t = 0$, half of the cells were treated with CHX (50 μg/mL) and both GFP and mCherry fluorescence were recorded in real-time for 8 h in a temperature and $CO_2$ controlled plate reader (CLARIOstar, BMG). GFP:mCherry ratios were calculated and values in CHX treated cells expressed relative to untreated cells.

### Statistics and reproducibility
No statistical method was used to predetermine sample size. No data were excluded from the analyses. The experiments were not randomised. The Investigators were not blinded to allocation during experiments and outcome assessment. Statistical tests were conducted as described with each Figure legend using GraphPad Prism v10.

### Reporting summary
Further information on research design is available in the Nature Portfolio Reporting Summary linked to this article.

## Data availability
The authors confirm that data supporting the findings presented in this study are available in the article, its supplemental data files and source data folder. Mass spectrometry proteomics data have been deposited to the ProteomeXchange Consortium (http://proteomecentral.proteomexchange.org) via the PRIDE partner repository (https://www.ebi.ac.uk/pride/) with the following dataset identifiers: PXD047612 (N-terminal acetylation of Human JunB in yeast expressing all N-terminal acetyltransferases). PXD045695 (LC-MSMS analysis of immunoprecipitated JunB-FLAG from HAP1 ADO KO cells). Source data are provided with this article. Source data are provided with this paper.

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

## Acknowledgements

K.H. was supported by the Oxford-GSK-Crick Doctoral Programme in Chemical Biology (Engineering and Physical Sciences Research Council grant EP/S513866/1, GlaxoSmithKline and The Francis Crick Institute). In addition, this work was supported by the Wellcome Trust (grant no. 106241/Z/14/Z) and the Ludwig Institute for Cancer Research (T.K. and P.J.R.) and by grants from the Research Council of Norway (RCN) (FRI-PRO Grant 324195 to T.A.), the Norwegian Cancer Society (171752-PR-2009-0222 to T.A.), and the European Research Council (ERC) under the European Union Horizon 2020 Research and Innovation Program (Grants 864888 to E.F. and 772039 to T.A.). Mass spectrometry-based proteomic analyses were performed by the Proteomics Unit at the University of Bergen (PROBE). This facility is a member of the National Network of Advanced Proteomics Infrastructure (NAPI), which is funded by the Research Council of Norway (INFRASTRUKTUR-program project number: 295910).

## Author contributions

K.H., T.K., N.Masson, P.J.R., T.A. and E.F. conceived and designed the study. K.H., T.K., M.M., M.L., N.McTiernan and S.A. conducted experiments. All authors analysed data and prepared figures. K.H., T.K., P.J.R., T.A. and E.F. wrote the manuscript with input from all authors.

## Competing interests

The authors declare no competing interests.
