## [Peer Review File · Nature Communications]

REVIEWER COMMENTS

Reviewer #1 (Remarks to the Author):

Comments to the authors:

The manuscript “N-terminal cysteine acetylation and oxidation patterns may define protein stability” uncovers that proteins displaying cysteines at their N-terminus can be subject to 2-aminoethanethiol dioxygenase (ADO) or N-terminal acetyltransferase A (NatA) depending on the sequence context. The authors combined *in vitro* and *in vivo* approaches to unambiguously demonstrate the determinants for recognizing the N-terminus by either ADO or NatA. Both enzymes differ substantially in their substrate preference, with only slightly overlapping substrate patterns. Noticeably, N-terminal acetylation of ADO substrates inhibits their recognition by ADO, strongly suggesting that co-translational acetylation of N-terminal cysteine residues may contribute to substrate recognition. Furthermore, the authors provide evidence that stimulus-triggered, and ADO-dependent cysteine oxidation destabilizes reporter constructs, including the 15mer potential N-degron. The authors could not prove that these N-degrons also act *in vivo* on full-length proteins. However, this is faithfully discussed and out of the focus of this initial study. The entire study is well-designed, and the results are logically ordered and presented. The manuscript is well-written and concise. The authors do not over-interpret the findings of their study and accentuate the shortcomings in their conclusion section. The findings are novel and add significant information to the field of oxygen sensing and N-terminal acetylation-dependent protein degradation in eukaryotes. This reviewer did not identify substantial experimental flaws but would recommend addressing the following concerns to strengthen the major claim and readability for non-expert readers:

Suggestions

Figure 4 The major claim of the manuscript would be strengthened if the RGS4 N-terminus (MCK) were converted to MCD/MCG and tested for stabilization of the dual-fluorescence oxygen reporter. The corresponding artificial RGS4 (MDG) protein could be expressed in cells with regular or decreased NatA activity under normoxic and hypoxic conditions. Alternatively to the latter experiment, the AAAS candidate (P 15) could be tested in this system.

Minor Concerns

- Figure 4B: Please re-order (on the X-axis) the analyzed DFORs according to their probability of being recognized by ADO and/or indicate that RGS4 and ASNS were included as references.
- P6 ...Nt-Cys initiating peptides representing residues 2-15 of MetMetCys-initiating proteins was measure....Please explain this selection to the non-expert reader.

- P8 .OS6, OS7 and OS10 sequences only differ at the residues following Nt-Cys, suggesting that small changes at this position... -> A statistical analysis in Fig. 3A is missing to support this suggestion.

- P8 ..we performed docking experiments... ->Please note right in the beginning that it is a molecular modeling approach.

- P17 ...One possible explanation for lack of in vivo regulation of these full-length proteins is protein folding. ->Discuss in this context that cell lines may not have expressed the relevant protein degradation system.

Reviewer #2 (Remarks to the Author):

This interesting and clearly presented manuscript set out to identify novel 2-aminoethanethiol dioxygenase (ADO) substrates that could be subject to O₂-regulated degradation via the N-degron pathway. The authors tested a subset of the 200-300 human MC-initiating proteins that are potential substrates. Importantly, they used both in vitro and cellular approaches.

Substrate preferences for ADO towards Cys initiating proteins were established by quantifying oxidation of synthetic peptides and supported by in silico docking experiments. Unsurprisingly however, not all in vitro substrates were subject to ADO-mediated degradation as judged by the in vivo dual-fluorescence oxygen reporter (DFOR) assay. The authors hypothesised that that other Nt modifications could preclude oxidation and through a variety of approaches found convincing evidence for Nt cysteine acetylation, additionally establishing some sequence preferences for NatA. Moreover, they provided evidence that Nt acetylation abrogates oxidation and vice versa.

Although the known specificity of methionine aminopeptidases implies that Nt cysteine is readily revealed co-translationally, acetylation of Nt cysteine residues has not previously been reported in any organism, to the best of my knowledge. Whilst the initial hypothesis was that Cys acetylation and Cys oxidation might be competing reactions at the same N-terminus, sequence preferences for residues downstream of Nt Cys dictate that many MC proteins are unlikely to be substrates for both enzymes and the authors concede that a relatively small number of proteins might be regulated in this way (I liked the honesty of the title "N-terminal cysteine acetylation and oxidation patterns _may_ define protein stability"). They found six proteins with acetylated Nt-Cys in publicly available mass spectrometry data from human cells. Are there examples from other organisms? Given the parallels between ADO and PCO it would be interesting if this modification is found in plants (I had a quick look but didn't find any examples).

It is proposed (Varshavsky 2019) that an N-degron comprises three elements: a destabilising residue at the N-terminus, one or more lysine residues in (3D) proximity to the N-terminus which itself must be accessible and not buried in a protein or protein complex. Related to this, whilst the DFOR assay provides a useful measure of protein stability, only the Nt sequences of candidate ADO substrates were used in this study. This is both a strength and a weakness: on one hand, it precludes any influence of other degrons in the candidate substrates, but on the other, the N-termini are not necessarily presented in their native 3D context. This was addressed by studying the stability of MC proteins in vivo and there is some discussion in the manuscript about shielding of degrons. Potentially, the manuscript could be improved by expanding this point with a possible structural rationale for the mismatch between in vitro and in vivo results. Are there crystal structures or AlphaFold models that would be informative?

Additional points:

The DFOR assay needs to be more clearly explained, e.g., does the GFP have lysines (for ubiquitination)? Is there a linker? How does the cleavage work?

Fig. 4B has three panels- are these separate experiments? The GFP:mCherry ratios for RGS4 and ASNS seem quite variable (see also Fig. 4C)- please could the authors comment on this? Would one expect the ratio to be one, where there is no ADO-catalysed degradation of the modified GFP?

It would be helpful for the non-specialist reader to mention other examples where competing post-translational modifications can alter the fate of a protein (e.g., acetylation and ubiquitination).

Reviewer #3 (Remarks to the Author):

In this manuscript the authors investigate a set of human proteins with Cys-2, as possible substrates of the oxygen-sensing protein ADO and downstream Arg/N-degron pathway. They show using short amino terminal peptides of a subset of these proteins in vitro that ADO has differential specificity based on sequence determinants close to Cys-2, allowing the design of artificial optimal and sub-optimal substrates. These are tested in vivo using an artificial assay system to assess the peptide sequence, that for the most part recapitulates the in vitro assay results. Interestingly the authors then show that substrates with very poor affinity with ADO can be targets for NatA acetylation. Finally, the authors assess oxygen and ADO-associated stability of Cys-2 full length proteins in cell culture.

The work is interesting because Cys-2 proteins have the potential to be components of a HIF-independent oxygen sensing system, and so defining the range of oxygen-sensitivity of Cys-2 proteins will greatly increase knowledge about the possible functions of this HIF-independent mechanism. Very few physiological Cys-2 substrates of ADO have as yet been identified and so experimental approaches important to define the scope of ADO biological function. The experiments are carried out to a high standard by researchers with a track record in the subject areas of both HIF and ADO biochemistry. The work moves the field forwards in defining the in vitro affinity of ADO for Cys-2 peptides, and in defining the role of Nat-A in acetylation of the amino terminus of these proteins. The authors postulate that acetylation or oxidation are two non-overlapping fates for Cys-2 proteins, and that the mutually exclusive affinities of ADO and Nat-A indicate independent pools of these substrates, though some peptides appear to have some mixed affinities. In this context Figure 6 represents a very nice overview of the in vitro analyses of the authors. Only around 10% of the total Cys-2 proteins in the human genome are investigated but there is no rationale given as to why these were chosen. The work mainly suffers from the incongruity between in vitro work with peptides and in vivo results assessing full length proteins with cognate antibodies (Figure 7). Peptides with very good affinity for ADO in vitro are shown when part of full length proteins not to be regulated by ADO/oxygen in cell cultures. This does not appear to provide confidence that initial in vitro approach of the authors is a valid method to identify ADO substrates, though only very few good ADO substrates were assessed in vivo. The authors may wish to comment on this. The observation that GTPF2 with only 8% of the relative ADO activity compared to RGS4 can be destabilised by ADO-OE suggests both that other proteins with higher relative activity could be investigated by the authors, and that perhaps other cell types with increased ADO activity might reveal other substrates. In addition, the determination of optimal substrate sequence downstream of Cys-2 (Fig. 2) could be followed by mutation of sequences of full length proteins to observe if this increases ADO/oxygen-regulated degradation in vivo. An assertion in the concluding section of the manuscript that Cys-2 may be hidden by protein folding in vivo also indicates that the in vitro approach is not fruitful. The authors assert that proteins in vivo may be substrates of ADO but not of the Arg/N-degron pathway, how would they envisage that that could be possible? Finally I missed more discussion by the authors about the relative potential of substrates for mutually exclusive regulation via NatA or ADO, beyond the statement that it is unlikely to be a common phenomenon. Changed stability of substrates with even quite low ADO affinity by acetylation might strongly alter function, do any of the known functions of the proteins investigated hint that this might be a possibility?

Figure 1 concerns N-terminal Cys proteins but (B) has an R group.

The authors state that 68 proteins were investigated but 71 proteins are presented in Table S2.

Figure 5, HMGA1 has a red asterisk presumably because it is a positive control, this is not highlighted in the figure legend.

Figure S6; should say (Fig. 5E). I could not find that this figure was referred to in the main text.

Figure 7C where is STAC2? Referred to in text and figure legend but not in figure.

Table S8; what does MCK stand for, not referred to in the main text.

RESPONSE TO REVIEWERS

We are very grateful for all of the Reviewers' positive and constructive comments. In light of the queries and suggestions provided, we have added further experimental results and addressed all the points raised, resulting in what we believe to be an improved and robust manuscript. A description of the changes we have made, as well as specific responses to the Reviewers' concerns, is given below. Please review these changes along with document 'NCOMMS-23-38359_R1_HIGHLIGHTED_CHANGES'.

In addition to addressing Reviewer's comments, we have included new data in the manuscript that further supports that *in vivo* acetylation of Nt-Cys is exclusively catalysed by NatA; these experiments were conducted in *S.cerevisiae*, which allowed complete knockout of *naa10* and *naa20*, and are presented in Figure 5 (also described in page 12, line 227 onwards). As a consequence, we have slightly rearranged the manuscript to separate these data from *in vitro* NatA-catalysed acetylation data which are now presented in Figure 6. Subsequent Figures have been re-numbered. All page numbers, line numbers and Figure numbers below refer to the revised manuscript.

Reviewer 1

The manuscript "N-terminal cysteine acetylation and oxidation patterns may define protein stability" uncovers that proteins displaying cysteines at their N-terminus can be subject to 2-aminoethanethiol dioxygenase (ADO) or N-terminal acetyltransferase A (NatA) depending on the sequence context. The authors combined in vitro and in vivo approaches to unambiguously demonstrate the determinants for recognizing the N-terminus by either ADO or NatA. Both enzymes differ substantially in their substrate preference, with only slightly overlapping substrate patterns. Noticeably, N-terminal acetylation of ADO substrates inhibits their recognition by ADO, strongly suggesting that co-translational acetylation of N-terminal cysteine residues may contribute to substrate recognition. Furthermore, the authors provide evidence that stimulus-triggered, and ADO-dependent cysteine oxidation destabilizes reporter constructs, including the 15mer potential N-degron. The authors could not prove that these N-degrons also act in vivo on full-length proteins. However, this is faithfully discussed and out of the focus of this initial study. The entire study is well-designed, and the results are logically ordered and presented. The manuscript is well-written and concise. The authors do not over-interpret the findings of their study and accentuate the shortcomings in their conclusion section. The findings are novel and add significant information to the field of oxygen sensing and N-terminal acetylation-dependent protein degradation in eukaryotes. This reviewer did not identify substantial experimental flaws but would recommend addressing the following concerns to strengthen the major claim and readability for non-expert readers:

Suggestions:

Figure 4 The major claim of the manuscript would be strengthened if the RGS4 N-terminus (MCK) were converted to MCD/MCG and tested for stabilization of the dual-fluorescence oxygen reporter. The corresponding artificial RGS4 (MDG) protein could be expressed in cells with regular or decreased NatA activity under normoxic and hypoxic conditions. Alternatively to the latter experiment, the AAAS candidate (P 15) could be tested in this system.

We thank the Reviewer for this helpful suggestion. We have now conducted peptide assays with CD and CG- initiating forms of RGS4 (RGS4 K3D and K3G, respectively) and find that, as anticipated, they are not good substrates for ADO (Figure 3B, page 8 lines 110-112 & 122-124, Supplementary Figure S3A). We also find that RGS4 K3D does not trigger destabilisation of GFP in the DFOR assay (Figure 4, pages 10/11, lines 167 & 192), consistent with the peptide results. We conducted an additional experiment to determine whether the stability of RGS4₁₋₁₅[K3D]-DFOR was impacted by hypoxia (Figure X1). Given that this peptide

was unable to destabilise GFP in normoxia, we were not surprised to find that there was no difference in stability of this construct in hypoxia; we therefore did not include this data in the main manuscript.

Figure X1. DFOR constructs were generated for RGS4 and RGS4 K3D and the stability of the GFP portion assessed in response to treatment with 2,2DIP (200 μ M) or hypoxia (1% O₂) for 24h. ASNS₁₋₁₅-DFOR was used as a negative control. Histograms represent the mean \pm S.D. from 3 independent biological replicates, and significance tested using a 2-way ANOVA with Dunnett's post-hoc, **P<0.01.

When we tested the RGS4 K3D peptide in the NatA *in vitro* assay (Figure 6B, page 13 lines 267-269, page 14 lines 290-292), we found that it led to an increase in NatA activity compared to RGS4 (6.0 and 1.2 % of HGMA1 control, respectively). Overall, the experiments confirm that the removal of the lysine at position 3 of RGS4 had a negative impact on ADO activity and its replacement with an aspartate had a positive effect on NatA activity. As suggested by the Reviewer, we also used siRNA to knock down *NAA10* (NatA) in SH-SY5Y cells stably expressing RGS4₁₋₁₅[K3D]-DFOR; following cycloheximide treatment we found RGS4₁₋₁₅[K3D]-GFP stability to be stable under these conditions (Supplementary Figure S8A, page 14 lines 292-295). This indicates that NatA knockdown has no impact on stability and RGS4 K3D stability arises from reduced ADO activity. However, it should be noted that it proved challenging to fully knockdown NatA therefore it is possible that low levels of Nt-Cys acetylation contributed to RGS4 K3D stabilisation. Equivalent experiments were conducted for the 'better' NatA substrates JunB and AAAS, examining their stability by Western blot. Similar to RGS4 K3D, siNAA10-mediated NatA knockdown did not result in any significant impact on JunB or AAAS stability in either WT or ADO knockout cells, suggesting that their lack of oxidation by ADO may dictate their stability rather than shielding of oxidation by acetylation. Again, however, incomplete knockdown of acetylation activity in these cells means it is hard to draw definitive conclusions; it is possible that acetylation of AAAS can prevent its oxidation by ADO (it is a weak ADO substrate *in vitro*) as proposed in page 16 (lines 335-338). We have included these data as a new Supplementary Figure (Figure S8) and refer to them on pages 14/15 (lines 292-297) and page 16-17 (lines 349-354).

Minor Concerns

- 1. Figure 4B: Please re-order (on the X-axis) the analyzed DFORs according to their probability of being recognized by ADO and/or indicate that RGS4 and ASNS were included as references.**

In recognition of this and a suggestion from Reviewer 2, we apologise that Figure 4B was not presented more clearly in our submitted manuscript. In the initial submitted version, we had presented this data as 3 separately conducted sets of experiments (with different gain settings dependent on overall fluorescence levels), each with RGS4 as a positive control and ASNS as a negative control. To present each experiment along a single x-axis, and therefore allow for easier comparison, we have normalised each outcome to its respective RGS4 + Dipyridyl control. In combination with the negative control construct, ASNS₍₁₋₁₅₎DFOR, this allows for appropriate inter-assay comparisons. We have also re-ordered the DFORs on the x-axis according to their probability of being recognised by ADO (based on outcome in peptide experiments, Figures 2/3). This reformatted data can be seen in a new version of Figure 4, where we have noted in the Legend that GFP:mCherry ratios have been normalised. Furthermore, we have added justification for normalisation to the relevant Materials and Methods section (page 23, lines 518-520).

- 2. P6 ...Nt-Cys initiating peptides representing residues 2-15 of MetMetCys-initiating proteins was measure....Please explain this selection to the non-expert reader.**

It is known that MetCys initiating proteins have their N-terminal Met residues co-translationally excised by Methionine AminoPeptidases¹. When selecting potential ADO substrates, we therefore chose proteins with N-terminal Met₁Cys₂- sequences but used corresponding peptide substrates for *in vitro* assays in which the initiating Met was not present. Peptides (14-mers) therefore had Cys₂ at their N-termini. We have clarified this on page 6, lines 79, by reminding readers that peptides represented 'residues 2-15 of MetCys-initiating proteins (from which iMet residues have been excised by MetAPs²⁰).'

3. ***P8 .OS6, OS7 and OS10 sequences only differ at the residues following Nt-Cys, suggesting that small changes at this position... -> A statistical analysis in Fig. 3A is missing to support this suggestion.***

Thank you for highlighting this omission. We have conducted statistical analyses to determine significant differences in ADO activity towards OS and AS peptides as shown in revised Figure 3. Statistically significant differences are summarised on the graphs (described in Figure 3 legend, page 37) and full analyses are available in Tables S3 and S4. The results of these analyses confirm that the rate of ADO activity towards OS6 is significantly different to its activity towards OS7 and OS10.

4. ***P8 ..we performed docking experiments... ->Please note right in the beginning that it is a molecular modeling approach.***

We have stated this clearly on page 9, line 137.

5. ***P17 ...One possible explanation for lack of in vivo regulation of these full-length proteins is protein folding. ->Discuss in this context that cell lines may not have expressed the relevant protein degradation system.***

We thank the Reviewer for this suggestion which we have considered carefully. We know from positive controls in our own experiments (RGS4, RGS5, Figure 8A), our own previously published work² and published work from others³ that the components of the Cys/Arg branch of the N-degron pathway are present in the cell lines used in this study (SH-SY5Y, U87-MG and HEK293T). The reason for SUSD6 and ANKRD29 stability *in vivo* is therefore unlikely to be due to lack of this protein degradation system, which is the system we have been focussing on. It remains possible that the stability of SUSD6 and ANKRD29 is regulated by a different pathway, and indeed the Cys/Arg N-degron pathway has been linked to protein degradation via autophagy, however this is triggered by non-enzymatic Cys oxidation rather than ADO-catalysed Cys oxidation⁴. Given SUSD6 and ANKRD29 *can* be ADO targets (as shown using our peptide and DFOR-based assays) we consider it unlikely that they are stabilised in cells due to a lack of (Cys/Arg N-degron driven-) autophagy protein degradation machinery. We would therefore respectfully argue that protein conformation and access to SUSD6 and ANKRD29 N-termini is most likely to confound access by ADO or other N-degron pathway components; we have expanded this point in the discussion of *in vivo* results (page 18, lines 377-384).

Reviewer 2

This interesting and clearly presented manuscript set out to identify novel 2-aminoethanethiol dioxygenase (ADO) substrates that could be subject to O₂-regulated degradation via the N-degron pathway. The authors tested a subset of the 200-300 human MC-initiating proteins that are potential substrates. Importantly, they used both in vitro and cellular approaches.

Substrate preferences for ADO towards Cys initiating proteins were established by quantifying oxidation of synthetic peptides and supported by in silico docking experiments. Unsurprisingly however, not all in vitro substrates were subject to ADO-mediated degradation as judged by the in vivo dual-fluorescence oxygen reporter (DFOR) assay. The authors hypothesised that that other Nt modifications could preclude oxidation and through a variety of approaches found convincing evidence for Nt cysteine acetylation, additionally establishing some sequence preferences for NatA. Moreover, they provided evidence that Nt acetylation abrogates oxidation and vice versa.

Although the known specificity of methionine aminopeptidases implies that Nt cysteine is readily revealed co-translationally, acetylation of Nt cysteine residues has not previously been reported in any organism, to the best of my knowledge. Whilst the initial hypothesis was that Cys acetylation and Cys oxidation might be competing reactions at the same N-terminus, sequence preferences for residues downstream of Nt Cys dictate that many MC proteins are unlikely to be substrates for both enzymes and the authors concede that a relatively small number of proteins might be regulated in this way (I liked the honesty of the title “N-terminal cysteine acetylation and oxidation patterns _may_ define protein stability”). They found six proteins with acetylated Nt-Cys in publicly available mass spectrometry data from human cells. Are there examples from other organisms? Given the parallels between ADO and PCO it would be interesting if this modification is found in plants (I had a quick look but didn't find any examples).

We agree with the Reviewer that it would be very interesting if Nt-Cys proteins were acetylated/oxidised with similar degrees of post-Cys sequence dependence in other organisms as this would suggest a highly evolved mechanism of controlling protein stability. We looked for Nt-acetylation of Nt-Cys initiating proteins in plants and found one reported example, Avirulence- induced gene-like protein (At5G39720), with the N-terminal sequence (M)CSSDSLQH⁵. Although this is consistent with the substrate preferences we observed for *HsNatA*, we have not reported this in the manuscript for three reasons:

(a) We know that PCO-like enzymes from angiosperms, bryophytes and algae show different substrate preferences⁶, which suggests that evolution of substrate specificity occurred after divergence of the plant and animal lineages. It is therefore difficult to draw parallels between Nt-Cys modification patterns between plants and animals.

(b) Limited data is available for plant Nt-Cys modifications therefore it is hard to draw firm conclusions about substrate preferences.

(c) We wish to maintain the focus of our study on human Nt-Cys modifications and considered that discussion of Nt-Cys acetylation in other organisms may complicate the manuscript.

We will nevertheless direct future research efforts to investigate whether a similar phenomenon of mutually exclusive acetylation/oxidation by impact protein stability in plants as this would be an area of significant interest in the plant proteostasis field.

It is proposed (Varshavsky 2019) that an N-degron comprises three elements: a destabilising residue at the N-terminus, one or more lysine residues in (3D) proximity to the N-terminus which itself must be accessible and not buried in a protein or protein complex. Related to this, whilst the DFOR assay provides a useful measure of protein stability, only the Nt sequences of candidate ADO substrates were used in this study. This is both a strength and a weakness: on one hand, it precludes any influence of other degrons in the candidate substrates, but on the other, the N-termini are not necessarily presented in their native 3D context. This was addressed by studying the stability of MC proteins in vivo and there is some discussion in the manuscript about shielding of degrons. Potentially, the manuscript could be improved by expanding this point with a possible structural rationale for the mismatch between in vitro and in vivo results. Are there crystal structures or AlphaFold models that would be informative?

We thank the Reviewer for this suggestion; we have indeed investigated whether there is structural information for SUSP6 and ANKRD29 which could rationalise their stability *in vivo* despite the ability of

their N-termini to promote ADO-dependent degradation in the DFOR and peptide assays. According to the RCSB Protein DataBank, there are no reported structures of either protein. AlphaFoldv2 has, however, been used to predict structures for both proteins. SUS6 is predicted to be a membrane-spanning protein with unfolded N- and C-termini. Although the N-terminus was predicted to be accessible, the confidence level in this area of the structure was low. SUS6 may be inaccessible due to subcellular localisation of its N-terminus being incompatible with that of ADO, or indeed accessibility of the N-terminus may be hindered due to protein folding or interaction with other proteins. AlphaFoldv2 predicts that ANKRD29 has an ankyrin repeat^{7,8} structure, with short exposed N- and C-termini, albeit with low confidence for the termini structure predictions. Ankyrin repeat proteins are typically involved in protein:protein interactions, therefore the chances of inaccessibility of the N-terminus are reasonably high and likely hinder access of ADO or other N-degron pathway components.

Given the high level of speculation, we have not discussed these suggestions extensively in the main text, but have expanded and clarified this part of the manuscript to indicate that structure predictions have informed our suggested rationale for SUS6 and ANKRD29 stability. We believe these comprise (i) inaccessibility due to protein folding, (ii) inaccessibility due to protein:protein interactions or (iii) inaccessibility due to incompatibility of subcellular localisation. We have also noted (as discussed in response to Reviewer 1 comment 5) that stability of SUS6 and ANKRD29 may also be due to inaccessibility of their N-termini to other components of the N-degron system, and/or ADO. Please see modified text on page 18, lines 377 – 386.

Additional Points:

The DFOR assay needs to be more clearly explained, e.g., does the GFP have lysines (for ubiquitination)? Is there a linker? How does the cleavage work?

We have included additional explanation for the DFOR assay in the legend of Figure S5 and made additional reference in the main text (page 10, line 166) to both this Figure and a Methods in Molecular Biology chapter where this assay is described in full⁹.

Fig. 4B has three panels- are these separate experiments? The GFP:mCherry ratios for RGS4 and ASNS seem quite variable (see also Fig. 4C)- please could the authors comment on this? Would one expect the ratio to be one, where there is no ADO-catalysed degradation of the modified GFP?

The Reviewer is correct, each panel in 4B represented a separate set of experiments. Gain settings required adjusting to accommodate the higher fluorescence values detected from some constructs (AAAS, JunB, GPX1, MTPN) in later experiments. This is why the GFP:mCherry ratios varied for the controls (RGS4 and ASNS). Concurrent with a re-arrangement of this data in response to Reviewer 1 (minor comment #1), we have chosen to normalise all experiments to their respective positive control (RGS4 + Dipyrindyl), thereby allowing all experiments to be presented on one graph.

Regarding the expectation of a GFP:mCherry to be one in the absence of degradation; in theory this is correct but differences in the inherent fluorescence properties of GFP and mCherry, and their detection parameters, mean this does not manifest in practice.

It would be helpful for the non-specialist reader to mention other examples where competing post-translational modifications can alter the fate of a protein (e.g., acetylation and ubiquitination).

Thank you for this suggestion, we have included the known examples of competitive acetylation and ubiquitin on Lys residues and competitive phosphorylation and O-GlcNAcylation on Ser/Thr residues (page 5 lines 47-50) to provide context for the reader that such competitive modifications on the same residue have previously been reported. We included this in the Introduction to provide context for our hypothesis that acetylation may have been impacting the ability of ADO to access Nt-Cys, however ultimately NtA

and ADO likely do not compete (at least on many proteins) for the same Nt-Cys residues given their post-Cys substrate preferences.

Reviewer 3

In this manuscript the authors investigate a set of human proteins with Cys-2, as possible substrates of the oxygen-sensing protein ADO and downstream Arg/N-degron pathway. They show using short amino terminal peptides of a subset of these proteins in vitro that ADO has differential specificity based on sequence determinants close to Cys-2, allowing the design of artificial optimal and sub-optimal substrates. These are tested in vivo using an artificial assay system to assess the peptide sequence, that for the most part recapitulates the in vitro assay results. Interestingly the authors then show that substrates with very poor affinity with ADO can be targets for NatA acetylation. Finally, the authors assess oxygen and ADO-associated stability of Cys-2 full length proteins in cell culture.

The work is interesting because Cys-2 proteins have the potential to be components of a HIF-independent oxygen sensing system, and so defining the range of oxygen-sensitivity of Cys-2 proteins will greatly increase knowledge about the possible functions of this HIF-independent mechanism. Very few physiological Cys-2 substrates of ADO have as yet been identified and so experimental approaches important to define the scope of ADO biological function. The experiments are carried out to a high standard by researchers with a track record in the subject areas of both HIF and ADO biochemistry. The work moves the field forwards in defining the in vitro affinity of ADO for Cys-2 peptides, and in defining the role of Nat-A in acetylation of the amino terminus of these proteins. The authors postulate that acetylation or oxidation are two non-overlapping fates for Cys-2 proteins, and that the mutually exclusive affinities of ADO and Nat-A indicate independent pools of these substrates, though some peptides appear to have some mixed affinities. In this context Figure 6 represents a very nice overview of the in vitro analyses of the authors. Only around 10% of the total Cys-2 proteins in the human genome are investigated but there is no rationale given as to why these were chosen.

The Cys-2 proteins selected to test as ADO substrates were chosen to represent a diversity of MetCys-initiating proteins in terms of the sequence following Cys-2, in order to determine the substrate specificity of ADO. We also included some proteins from the human proteome whose N-terminal sequences resembled those of known substrates RGS4/5 and IL32. We have amended our justification (page 7 line 82-84) to clarify our rationale for selection.

The work mainly suffers from the incongruity between in vitro work with peptides and in vivo results assessing full length proteins with cognate antibodies (Figure 7). Peptides with very good affinity for ADO in vitro are shown when part of full length proteins not to be regulated by ADO/oxygen in cell cultures. This does not appear to provide confidence that initial in vitro approach of the authors is a valid method to identify ADO substrates, though only very few good ADO substrates were assessed in vivo. The authors may wish to comment on this.

The Reviewer is correct that some of the peptides which appeared to be good ADO substrates in the *in vitro* assay were not good substrates in the biological context. However, we would argue that the *in vitro* assay for ADO activity towards peptide substrates is a valid method. By studying ADO-peptide activity in isolation, the assay reveals which proteins *have the potential to be* regulated by ADO *in vivo*. This assay revealed, in a relatively high-throughput manner, that ADO activity is not promiscuous towards all Nt-Cys initiating substrates and instead showed a preference for peptides with basic or aromatic residues following the Cys. We have added a statement to reinforce this point in the main text: 'These sequences were selected to represent a diversity... which could provide information on ADO's substrate preferences and the potential of their corresponding proteins to be regulated by ADO *in vivo*' (page 7, line 84-85).

The observation that GTPF2 with only 8% of the relative ADO activity compared to RGS4 can be destabilised by ADO-OE suggests both that other proteins with higher relative activity could be investigated by the authors, and that perhaps other cell types with increased ADO activity might reveal other substrates.

We agree that there is more work to be done to identify novel ADO substrates and that our study has not pursued every potential ADO substrate identified through the peptide screens in the *in vivo* assays. Rather, this manuscript focussed on the capability of Nt-Cys initiating substrates to be divergently regulated by ADO or NatA in a sequence-dependent manner. GTPF2 was indeed shown to be regulated *in vivo* (albeit only when ADO was overexpressed, page 17 line 359-362); we included these data to illustrate the point that *in vitro* and *in vivo* data may not always be directly consistent, including that 'poor' substrates *in vitro* might be 'better' substrates *in vivo*. However, we do highlight that GFPT2 only becomes a substrate *in vivo* under artificial conditions of overexpression. We have included a comment to highlight the fact that other substrates from Figure 2 may yet prove to be ADO-regulated *in vivo* (page 17, line 363-365).

In addition, the determination of optimal substrate sequence downstream of Cys-2 (Fig. 2) could be followed by mutation of sequences of full length proteins to observe if this increases ADO/oxygen-regulated degradation in vivo.

This would indeed be an interesting experiment to conduct, but we believe we have addressed this point with the additional peptide and DFOR experiments addressing Reviewer 1's major point 1 (Figures 3, 4 and 6), exploring RGS4 K3D. We do not believe that the (long duration) equivalent experiment in full length protein is necessary to support the conclusions of this paper.

An assertion in the concluding section of the manuscript that Cys-2 may be hidden by protein folding in vivo also indicates that the in vitro approach is not fruitful.

We would respectfully argue that this is not the case. We acknowledge in the manuscript that the *in vitro* assays cannot mimic interaction of ADO with intact protein substrates, indeed our workflow (peptide assay, DFOR assay, endogenous protein assay) was designed to address this very issue. Nevertheless, as described above (Reviewer 3 point 2), the *in vitro* assay does demonstrate which proteins have the biophysical potential to be regulated by ADO. This data allows informed choices regarding which substrates should be pursued in more resource-intensive *in vivo* assays. Furthermore, the *in vitro* assays revealed the sequence components which indicate whether proteins are likely to be acetylated rather than oxidised, allowing us to identify the potential for Nt-Cys proteins to be alternately regulated by these two modifications.

The authors assert that proteins in vivo may be substrates of ADO but not of the Arg/N-degron pathway, how would they envisage that that could be possible?

For proteolytic degradation of ADO substrates *in vivo*, they must be (i) oxidised by ADO, (ii) arginylated by Arg transferase (ATE1) and (iii) ubiquitinated by E3 ligases (UBR1/2)⁴. It is possible that despite ADO-catalysed oxidation of Nt-Cys, the substrates are not recognised by ATE1 or UBR1/2 and therefore remain stable (e.g. substrate specificity dictated by residues Cys+1 has been reported for ATE1¹⁰ and UBR1 is known to target different subsets of proteins in a stress-signal dependent manner¹¹). We realise this point was not clear in the manuscript and have expanded this statement to clarify that we are referring to the additional ability of these enzymes to recognise ADO-oxidised substrates (page 18, line 384).

Finally I missed more discussion by the authors about the relative potential of substrates for mutually exclusive regulation via NatA or ADO, beyond the statement that it is unlikely to be a common phenomenon. Changed stability of substrates with even quite low ADO affinity by acetylation might strongly alter function, do any of the known functions of the proteins investigated hint that this might be a possibility?

We agree with the Reviewer that this is an interesting question. One example could be DYNLL1, whose stability has recently shown to be promoted by the lncRNA DLEU1 in esophageal squamous cell carcinoma cells via prevention of ubiquitination at K36 and K43¹², promoting cell survival/preventing apoptosis; it is possible that Nt-Cys acetylation of DYNLL1 has the same effect. We were reluctant to speculate on the dual regulation of specific proteins in the main text, in particular given the co-translational and irreversible

nature of NatA activity as understood to date. However we have added a statement to highlight the potential for dual regulation to impact stability of certain proteins (page 18, line 400-401).

Figure 1 concerns N-terminal Cys proteins but (B) has an R group.

We intended that Figure 1 illustrates the N-terminal modifications catalysed by ADO and NatA, which in the case of NatA is not limited to Nt-Cys residues. We have therefore retained the R group in (B) but altered the title of this Figure.

The authors state that 68 proteins were investigated but 71 proteins are presented in Table S2.

Table S2 includes 3 proteins that were already established as ADO substrates (RGS4/RGS5/IL32)²; we have adjusted the text on page 6 line 81 to state that ‘We selected 68 novel MC-initiating sequences from the human proteome to screen as potential ADO substrates’. Hopefully this improves clarity for the reader.

Figure 5, HMGA1 has a red asterisk presumably because it is a positive control, this is not highlighted in the figure legend.

Thank you for flagging this, the Reviewer is correct – we have noted that the positive control has a red asterisk in the Figure legend.

Figure S6; should say (Fig. 5E). I could not find that this figure was referred to in the main text.

Thank you for flagging this, the Reviewer is correct. We have amended Figure S6 title to include (Fig. 5C) [for updated Figure 5] and have included a reference to Fig. S6 in the main text page 12, line 234 and Figure 5C Figure legend.

Figure 7C where is STAC2? Referred to in text and figure legend but not in figure.

Thank you for flagging this, we had initially included a potential substrate called STAC2 but did not include it in the final version of this paper as the Nt-Cys initiating form is a non-canonical isoform. Apologies that we caused confusion by not removing every reference to it – we have now removed all reference to STAC2 in the text and figure legend.

Table S8; what does MCK stand for, not referred to in the main text.

MCX in Table S8 (now Table S10) refers to the constructs prepared for the DFOR assay (page 22, line 493). We have clarified this in the legend for Table S10.

References

1. Moerschell, R. P., Hosokawa, Y., Tsunasawa, S. & Sherman, F. The specificities of yeast methionine aminopeptidase and acetylation of amino-terminal methionine in vivo. Processing of altered iso-1-cytochromes c created by oligonucleotide transformation. *J. Biol. Chem.* **265**, 19638–19643 (1990).
2. Masson, N. *et al.* Conserved N-terminal cysteine dioxygenases transduce responses to hypoxia in animals and plants. *Science (80-.)*. **365**, 65–69 (2019).
3. Vu, T. T. M. & Varshavsky, A. The ATF3 Transcription Factor Is a Short-Lived Substrate of the Arg/N-Degron Pathway. *Biochemistry* **59**, 2796–2812 (2020).
4. Heo, A. J., Ji, C. H. & Kwon, Y. T. The Cys/N-degion pathway in the ubiquitin–proteasome system

and autophagy. *Trends Cell Biol.* **33**, 247–259 (2023).

5. Zhang, H. *et al.* N-terminomics reveals control of Arabidopsis seed storage proteins and proteases by the Arg/N-end rule pathway. *New Phytol.* **218**, 1106–1126 (2018).
6. Taylor-Kearney, L. J. *et al.* Plant Cysteine Oxidase Oxygen-Sensing Function Is Conserved in Early Land Plants and Algae. *ACS Bio Med Chem Au* **2**, 521–528 (2022).
7. Zhao, H. *et al.* ANKRD29, as a new prognostic and immunological biomarker of non-small cell lung cancer, inhibits cell growth and migration by regulating MAPK signaling pathway. *Biol. Direct* **18**, 28 (2023).
8. Sedgwick, S. G. & Smerdon, S. J. The ankyrin repeat: a diversity of interactions on a common structural framework. *Trends Biochem. Sci.* **24**, 311–316 (1999).
9. Smith, E. & Keeley, T. P. Monitoring ADO dependent proteolysis in cells using fluorescent reporter proteins. in *Methods in Enzymology* (Academic Press, 2023). doi:10.1016/bs.mie.2023.02.004
10. Wadas, B., Piatkov, K. I., Brower, C. S. & Varshavsky, A. Analyzing N-terminal arginylation through the use of peptide arrays and degradation assays. *J. Biol. Chem.* **291**, 20976–20992 (2016).
11. Szoradi, T. *et al.* SHRED Is a Regulatory Cascade that Reprograms Ubr1 Substrate Specificity for Enhanced Protein Quality Control during Stress. *Mol. Cell* **70**, 1025-1037.e5 (2018).
12. Li, Q. *et al.* DLEU1 promotes cell survival by preventing DYNLL1 degradation in esophageal squamous cell carcinoma. *J. Transl. Med.* **20**, 245 (2022).

REVIEWERS' COMMENTS

Reviewer #1 (Remarks to the Author):

The authors satisfactorily addressed all of my concerns. I have no further comments to improve the manuscript.

Reviewer #2 (Remarks to the Author):

I consider that the authors have addressed my comments in the revised manuscript and they have added additional experimentation to address comments of the other referees. Line 106 refers to Fig. 1B- do they mean Fig. 2B? Similarly, line 281 refers to Fig. 6F- presumably Fig. 6C?

Reviewer #3 (Remarks to the Author):

The authors have addressed the comments that I raised